# Metabolic shifts toward glutamine regulate tumor growth, invasion and bioenergetics in ovarian cancer

Lifeng Yang[1,2], Tyler Moss[3], Lingegowda S Mangala[4,5], Juan Marini[6], Hongyun Zhao[1,2], Stephen Wahlig[1,7], Guillermo Armaiz-Pena[4,5], Dahai Jiang[4,5], Abhinav Achreja[1,2], Julia Win[1,2], Rajesha Roopaimoole[4,5], Cristian Rodriguez-Aguayo[5,7], Imelda Mercado-Uribe[8], Gabriel Lopez-Berestein[5,7], Jinsong Liu[8], Takashi Tsukamoto[9], Anil K. Sood[4,5], Prahlad T Ram[3] & Deepak Nagrath[1,2,10,*]

## Abstract

Glutamine can play a critical role in cellular growth in multiple cancers. Glutamine-addicted cancer cells are dependent on glutamine for viability, and their metabolism is reprogrammed for glutamine utilization through the tricarboxylic acid (TCA) cycle. Here, we have uncovered a missing link between cancer invasiveness and glutamine dependence. Using isotope tracer and bioenergetic analysis, we found that low-invasive ovarian cancer (OVCA) cells are glutamine independent, whereas high-invasive OVCA cells are markedly glutamine dependent. Consistent with our findings, OVCA patients' microarray data suggest that glutaminolysis correlates with poor survival. Notably, the ratio of gene expression associated with glutamine anabolism versus catabolism has emerged as a novel biomarker for patient prognosis. Significantly, we found that glutamine regulates the activation of STAT3, a mediator of signaling pathways which regulates cancer hallmarks in invasive OVCA cells. Our findings suggest that a combined approach of targeting high-invasive OVCA cells by blocking glutamines entry into the TCA cycle, along with targeting low-invasive OVCA cells by inhibiting glutamine synthesis and STAT3 may lead to potential therapeutic approaches for treating OVCAs.

**Keywords** cancer metabolism; glutamine dependence; glutaminolysis; invasion; ovarian cancer

**Subject Categories** Metabolism; Cancer

**Mol Syst Biol. (2014) 10: 728**

## Introduction

Most ovarian cancers (OVCAs) are detected when metastasis has already occurred, often to the peritoneum (Nieman *et al*, 2011; Armaiz-Pena *et al*, 2013; Arvizo *et al*, 2013). Therefore, considerable interest in the cancer field is focused on new insights into factors regulating cancer invasion and migration along with tumor growth (Klopp *et al*, 2012; Kolonin, 2012; Arvizo *et al*, 2013; Dragosavac *et al*, 2013). A richly diverse tumor microenvironment implies that cells need appropriate nutrient for growth, invasion and energy metabolism. A growing number of studies indicate that the metabolic dependencies of cancer cells are regulated by oncogenic alterations which rewire the cellular metabolism to promote tumorigenicity (Dang *et al*, 2009; Gao *et al*, 2009; Weinberg *et al*, 2010; Gaglio *et al*, 2011; Csibi *et al*, 2013; Son *et al*, 2013). However, from recent studies focused on cancer invasion and malignant tumors, it has become apparent that metabolic dependencies and tumor metabolism are also dependent on cancer stage (Nomura *et al*, 2010; Benjamin *et al*, 2012; Caneba *et al*, 2012; Agus *et al*, 2013; Kim & Wirtz, 2013). Thus, nutrient addiction in cancer needs to be considered in a novel light: a synergism of malignant state and oncogenic alterations. Both glucose (Glc) and glutamine (Gln) have been shown to support cancer cell growth; however, the linkage between cancer cells' invasive and migratory potential and metabolic dependencies are not well studied and thus less appreciated.

There is mounting evidence on the dynamic interactions of glucose metabolism with cancer cell's survival pathways implicating invasion, migration and energetic homeostasis (Vander Heiden *et al*, 2010; Locasale & Cantley, 2011; Graham *et al*, 2012; Komurov

---

1  Laboratory for Systems Biology of Human Diseases, Rice University, Houston, TX, USA
2  Department of Chemical and Biomolecular Engineering, Rice University, Houston, TX, USA
3  Department of Systems Biology, University of Texas MD Anderson Cancer Center, Houston, TX, USA
4  Departments of Gynecological Oncology and Cancer Biology, University of Texas MD Anderson Cancer Center, Houston, TX, USA
5  Center for RNA Interference and Non-Coding RNA, University of Texas MD Anderson Cancer Center, Houston, TX, USA
6  Baylor College of Medicine, Houston, TX, USA
7  Department of Experimental Therapeutics, University of Texas MD Anderson Cancer Center, Houston, TX, USA
8  Department of Pathology, University of Texas MD Anderson Cancer Center, Houston, TX, USA
9  Johns Hopkins University, Baltimore, MD, USA
10 Department of Bioengineering, Rice University, Houston, TX, USA
   *Corresponding author. Tel: +1 713 348 6408; Fax: +1 713 348 5478; E-mail: deepak.nagrath@rice.edu

*et al*, 2012). Many glycolysis targeting drugs modulate cellular differentiation, antiapoptotic response and metastasis (Turkson & Jove, 2000; Levy & Lee, 2002; Schindler *et al*, 2007) through activation pathways which include epidermal growth factor receptor (EGFR) kinase, Src, Janus-activated kinase (JAK) and extracellular signal-regulated kinase (ERK; Chung *et al*, 1997; Garcia *et al*, 1997, 2001; Yu *et al*, 2003; Schindler *et al*, 2007). The linkage between Gln metabolism and these cancer cellular networks, which control cancer phenotype, is unclear.

Recent studies have shown that cancer cells are 'Gln addicted', which means that exogenous Gln deprivation leads to cell death (Weinberg *et al*, 2010; Wise & Thompson, 2010; Le *et al*, 2012). Although Gln is normally regarded as a nonessential amino acid, it is treated as an essential amino acid in these Gln-addicted cancer cells. For example, Gln can serve as an anaplerosis metabolite to drive the tricarboxylic acid (TCA) cycle for energy generation, it can generate the antioxidant glutathione to remove reactive oxygen species (ROS) and can be converted into other nonessential amino acids (NEAAs), purines and pyrimidines, and fatty acids for cellular replication (Johnson *et al*, 2003; Gaglio *et al*, 2009; Dang, 2010; DeBerardinis & Cheng, 2010; Weinberg *et al*, 2010; Wise & Thompson, 2010; Rajagopalan & DeBerardinis, 2011; Shanware *et al*, 2011; Daye & Wellen, 2012). Gln can activate cellular signaling, like the mTOR pathway through a bidirectional transport mechanism. Through these pathways, Gln regulates cellular protein translation, cell growth and autophagy (Gaglio *et al*, 2009; Nicklin *et al*, 2009). Despite the discovery of the essential role of Gln in tumorigenesis, it is not known whether Gln dependence in cancer cells is correlated with cancer aggressiveness—defined as cells with abnormally high potential for migration and invasion (Nomura *et al*, 2010). The differential rewiring of Gln metabolic pathways in low- and high-invasive OVCA cells may not only support survival, fuel cell growth, increase tumorigenesis and shift energy metabolism, but also increase migration and invasion, thereby increasing the metastatic potential of cancer cells. Thus, there is a pressing need to study the linkage between Gln metabolism and cancer invasiveness.

Here, we report a novel mechanism by which cancer cells maintain their invasiveness. We reveal that Gln, rather than Glc, may play a critical role in high-invasive ovarian tumor growth and in overall patient survival. Our data show that dependence on Gln in cancer cells is strongly correlated with cancer invasiveness. Low-invasive cells are found to be Gln independent, whereas high-invasive cells are extremely Gln dependent. We found significant correlation between patient survival and gene expression in the glutaminolysis/glycolytic metabolic network; genes involved in glutaminolysis and TCA cycle metabolic pathways are highly expressed in patients with poor survival, whereas glycolytic genes are associated with better patient survival. Using isotope tracers and bioenergetics, we uncover Gln's differential modulation of cancer cell hallmarks including deregulated cellular energetics (Dang, 2010; Hanahan & Weinberg, 2011; Koppenol *et al*, 2011), reductive carboxylation for biosynthesis (Metallo *et al*, 2012; Mullen *et al*, 2012) and shifts in energy metabolism (Vander Heiden *et al*, 2009; Sandulache *et al*, 2011; Liu *et al*, 2013), between low- and high-invasive cancer cells. Moreover, we show that Gln orchestrates cell survival, cancer progression and rewiring of metabolic pathways in high-invasive cancer cells by interacting with cellular networks that regulate homeostasis and cellular functions. We identified that Gln

withdrawal selectively reduces phosphorylated STAT3 expression in high-invasive cancer cells. Furthermore, forced expression of STAT3 in Gln-deprived cells can abrogate Gln-induced cell death. Our study uncovers previously undescribed roles of Gln in low-invasive versus high-invasive OVCA cells. This may provide avenues for targeting cancer cells not only in a stage-specific manner, but also by treating heterogeneous populations uniformly with drug cocktails to target mutually compensatory metabolic pathways in tumors.

# Results

## Glutamine dependence in ovarian cancer cells is correlated with cancer invasiveness and patient survival

To understand Gln dependence in OVCAs cells, we first analyzed cellular proliferation in an array of eight OVCA cell lines under complete medium, Gln-deprived and Glc-deprived conditions. We found that OVCAR3, IGROV1 and OVCA429 are Gln independent, OVCAR8 and OVCA420 are moderately Gln dependent, and SKOV3, SKOV3ip and Hey8 are highly Gln dependent (Fig 1A). Consistent with cellular proliferation in normoxia, OVCAR3 and SKOV3 showed similar behavior under hypoxia, where OVCAR3 is Gln independent and SKOV3 is Gln dependent (Supplementary Fig S1A). However, growth in all OVCA cell lines is Glc dependent (Fig 1B). Clonogenic growth experiments confirmed that Gln is critical for SKOV3 cell growth but has no influence on IGROV1 OVCA cells (Fig 1C). To uncover the linkage between Gln dependence and cellular invasion and migration, we performed Transwell invasion and wound-healing assays, respectively. Interestingly, we found that Gln dependence is correlated with the invasion and migratory capacity of both cancer cell lines (Fig 1D and E, Supplementary Fig S1B). Gln-independent OVCA cell lines OVCAR3, IGROV1 and OVCA429 are low invasive; highly Gln-dependent OVCA cell lines SKOV3, SKOV3ip and Hey8 are high invasive, while OVCAR8 and OVCA420 display intermediate behavior (Fig 1D and E). We did not find differential effects of Glc deprivation with respect to invasiveness in OVCA cell lines (Supplementary Fig S1C). Further, the expression of cell cycle proteins, specifically CDC2 and cyclin D1, correlates with cell proliferation levels in both Gln and Glc deprivation conditions (Fig 1F). Following Glc depletion, phosphorylated CDC2 decreases only in low-invasive OVCA cells (OVCAR3) and moderately invasive OVCA cells (OVCA420; Fig 1F, Supplementary Fig S1D). The high-invasive OVCA cells (SKOV3 and SKOV3ip) respond negatively whenever Glc or Gln is depleted from the culturing medium. Notably, cyclin D1 decreases only under Gln deprivation in high-invasive OVCA cells. Furthermore, loss of E-cadherin expression in high-invasive OVCA cells is consistent with their higher metastatic potential (Fig 1D). These results present evidence that high-invasive OVCA cells are Gln addicted, whereas low-invasive OVCA cells are Gln independent.

To determine whether the Gln-dependent phenotype of high-invasive OVCA cells in our cell lines models has clinical significance, we analyzed gene expression and survival data from OVCA patients within The Cancer Genome Atlas (TCGA). We built a network of the glucose and Gln metabolism within the TCA cycle and calculated the Cox regression values for each of the enzymes in the network to determine how expression levels correlate with

survival (Komurov *et al*, 2012). The correlation of metabolic network genes with OVCA patient survival shows poor outcome in patients with a higher expression level of genes involved in glutaminolysis, such as glutaminase (GLS1), glutamate dehydrogenase (GLUD1) and glutamate oxaloacetate transaminase (GOT1, GOT2), as well as genes involved in TCA cycle metabolic pathways and pyruvate conversion to acetyl-CoA like pyruvate dehydrogenase (PDH), citrate synthetase (CS), aconitase (ACO2) and succinate dehydrogenase B (SDHB; Fig 1G). This is in contrast to genes involved in Glc uptake and entry into the TCA cycle. These data

suggest that increased glutaminolysis supports mitochondrial TCA cycle activity in patients with high-invasive OVCA, resulting in the observation that glutaminolysis rather than glycolysis correlates with poor patient survival. Our findings are in contrast to previous studies that show correlation of poor outcome with high expression of glycolysis genes in some cancers (Haber *et al*, 1998; Kunkel *et al*, 2003). To confirm whether our findings are specific to OVCA, we analyzed the TCGA data from breast and kidney cancer patients. The gene expression network in kidney cancer patients shows that high expression of Glc metabolism genes correlates with poor

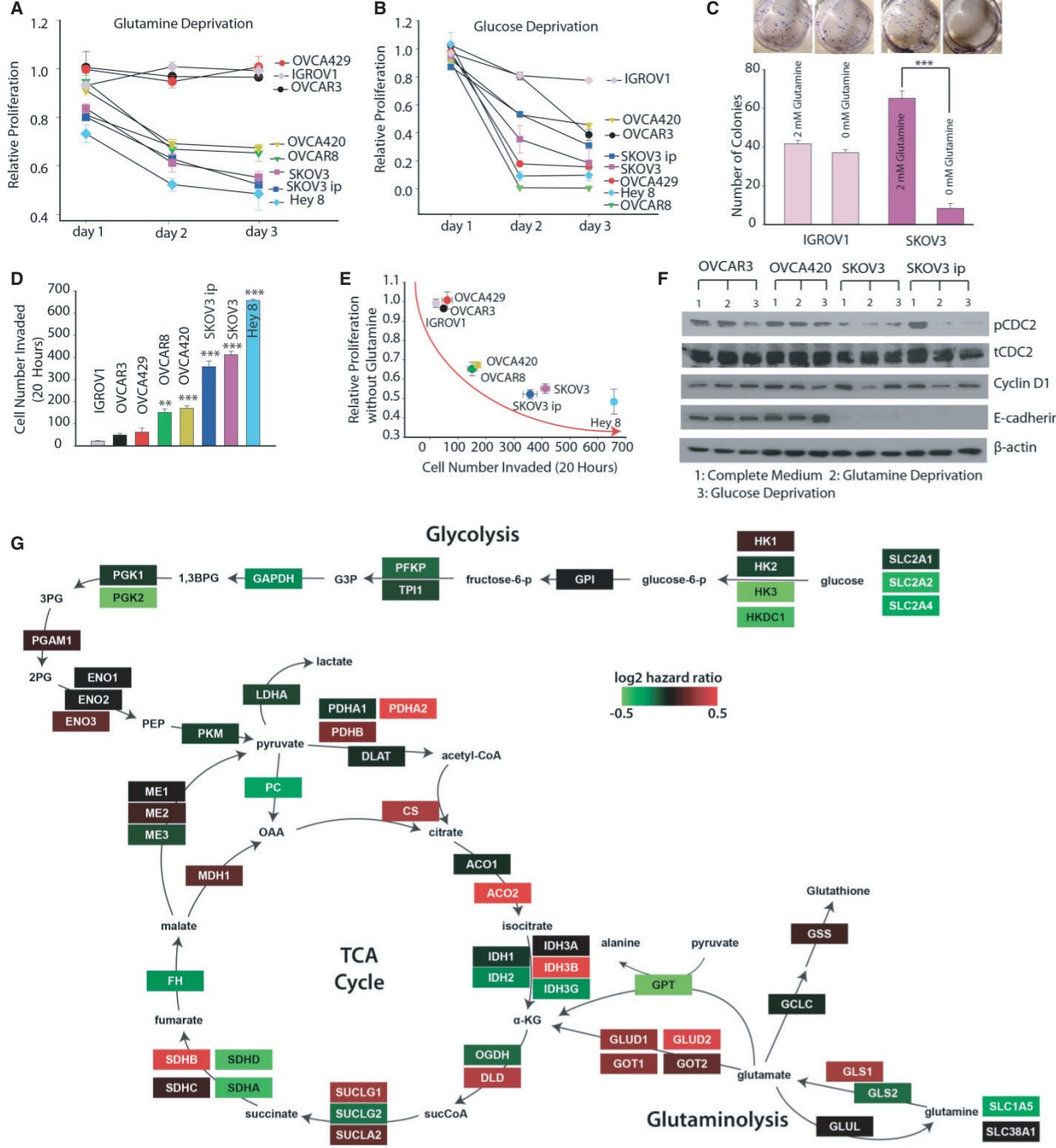

**Figure 1**

survival, while in breast cancer patients TCGA data show that both Glc and Gln metabolism correlate with poor survival (Supplementary Fig S2A and B).

## Glutamine differentially increases TCA cycle metabolite abundances in high-invasive versus low-invasive ovarian cancer cells

Recently, we identified that during migration, high- and low-invasive OVCA cells exhibit differing consumption of pyruvate. This discrepancy may be due to differential metabolite pools available in cancer cells (Caneba *et al*, 2012). Here, we asked whether high- and low-invasive OVCA cells have different metabolite abundances. Since both Glc and Gln can contribute to TCA cycle metabolites, we also questioned whether high- and low-invasive OVCA cells utilize different sources (Glc or Gln) to maintain their metabolite pool. To answer this, we performed $^{13}$C GC-MS-based isotope tracer analysis using labeled U-$^{13}$C$_5$ Gln and/or a 1:1 mixture of U-$^{13}$C$_6$ Glc and 1-$^{13}$C$_1$ Glc (Fig 2A and E). This analysis can reveal the contributions of a substrate within a particular metabolic pathway. For instance, a 1:1 mixture of U-$^{13}$C$_6$ Glc and 1-$^{13}$C-labeled Glc will provide both M3 (three $^{13}$C-labeled carbons) and M1 (one $^{13}$C-labeled carbon) pyruvate. Furthermore, M3 pyruvate will be converted into M2 and M3 malate and fumarate (M2 is produced in the first complete TCA cycle and M3 is from the second cycle); whereas M1 pyruvate will result in M1 and M2 malate and fumarate (M1 is produced in the first complete TCA cycle and M2 is from the second cycle). Conversely, the unlabeled M0 malate and fumarate will be obtained mainly from non-Glc sources (i.e. Gln, etc.). Thus, the relative abundance of isotopic labels between different cell lines reveals the preferential utilization of Glc toward replenishment of the TCA cycle metabolite pool (Fig 2A).

The mass isotope distribution (MID) of malate and fumarate using labeled Glc indicated that OVCAR3, the low-invasive OVCA cell line, has a higher conversion rate of Glc into TCA cycle intermediates than SKOV3, the high-invasive OVCA cell line. Interestingly, using a mixture of U-$^{13}$C$_6$ Glc and 1-$^{13}$C$_1$ labeled Glc, we found that the percentage of M2 and M3 malate is significantly higher in low-invasive OVCA cells (OVCAR3), while that the percentage of M0 malate is significantly lower (Fig 2B). Similar results were obtained for fumarate (Fig 2C). In addition, we found that OVCA cells

convert Glc into glutamate and that this conversion flux is higher in the low-invasive OVCA cell line, OVCAR3 (Fig 2D).

The Gln contribution to TCA cycle metabolite pools was estimated using labeled U-$^{13}$C$_5$ Gln and its conversion to M5 glutamate, M4 fumarate, M4 malate and M4 citrate (Fig 2E). The relative isotope abundances between different cell lines reveal Gln utilization for TCA cycle metabolite pool. Interestingly, we found that high-invasive OVCA cells (SKOV3) use Gln to produce 40% of their total TCA intermediate pool, while Gln contributes to only 15% of the TCA intermediate pool for low-invasive OVCA cells (Fig 2F). These data support the hypothesis that high-invasive OVCA cells preferentially use Gln for their TCA cycle metabolite pools, whereas low-invasive cells utilize Glc.

Recently, reductive carboxylation was found to promote tumor growth by supplying citrate for fatty acid synthesis (Metallo *et al*, 2012; Mullen *et al*, 2012). In this process, Gln, through an α-keto-glutarate (α-KG) intermediate, produces citrate for lipogenesis using isocitrate dehydrogenase 1/2 (IDH1/2). The rate of Gln-dependent reductive carboxylation is estimated by U-$^{13}$C$_5$ Gln-derived M5 citrate, M3 malate, M3 fumarate, M3 aspartate (Fig 2E). The high-invasive cancer cells (SKOV3) have at least twofold higher Gln-dependent reductive carboxylation flux compared to low-invasive cancer cells (OVCAR3; Fig 2G). In contrast, low-invasive OVCA cancer cells have higher pyruvate-dependent reductive carboxylation flux using pyruvate carboxylase (PC). Glc entry into the TCA cycle via pyruvate carboxylation involves an alternate pathway where pyruvate is directly converted to oxaloacetate (Supplementary Fig S3A). PC converts M3 pyruvate into M3 oxaloacetate, which condenses with acetyl-CoA to produce M3, M4 and M5 citrate (Supplementary Fig S3A). OVCAR3 has higher pyruvate carboxylation flux, compared to high-invasive cancer cells, to support its Gln independent growth (Cheng *et al*, 2011; Supplementary Fig S3B). Additionally, we found that the M4 malate and fumarate levels, obtained through pyruvate carboxylation, are significantly higher in low-invasive OVCA cells (OVCAR3; Fig 2B and C).

Next, we determined whether differential utilization of Gln and Glc in high- and low-invasive cancer cells could be the result of atypical metabolic reprogramming of these cells. We measured the oxygen consumption rate (OCR) and extracellular acidification rate (ECAR) of several OVCA cell lines. Surprisingly, we observed that the low-invasive OVCA cells (OVCAR3) have higher levels of basal

---

**Figure 1.   Glutamine addiction and cancer invasiveness are positively correlated in ovarian cancer cells (OVCA).**

A    Gln deprivation effect on proliferation rate of a panel of OVCA cells for 24, 48 and 72 h. OVCAR3, IGROV1, OVCA429 cells were Gln independent; OVCA420 and OVCAR8 cells were moderately Gln dependent; and SKOV3, SKOV3ip and Hey 8 cells were Gln dependent. $n \geq 15$.

B    Glucose deprivation effect on proliferation rate of OVCA cells for 24, 48, 72 h. $n \geq 15$.

C    Gln deprivation effect on clonogenic formation in IGROV1 and SKOV3. $n = 6$.

D    Matrigel invasion assay was conducted to characterize invasiveness of OVCA cell lines. OVCAR3, IGROV1 and OVCA429 cells were noninvasive, OVCA420 and OVCAR8 cells were moderately invasive, while SKOV3, SKOV3ip and Hey 8 were highly invasive OVCA cell lines. $n \geq 6$.

E    Correlation between proliferation rate at 72 h of OVCA cell lines under Gln-depleted conditions with their corresponding number of invaded cells.

F    Western blot analysis of cell cycle proteins linked with growth rate (CDC2, Cyclin D1) and protein linked with metastatic potential (E-cadherin) in OVCA cell lines. β-actin was used as the loading control.

G    Genes in the glutaminolysis and tricarboxylic acid cycle metabolic pathways are associated with higher risk in OVCA patients. Shown is a metabolic network and the genes (rectangles) that produce the enzyme catalyzing the reactions in the network. The genes in the network are colored by their correlation with OVCA patient survival based on calculating their Cox hazard values (color key). The gene expression and clinical data were fitted with a Cox proportional hazard model to determine the hazard ratio for each gene in Gln/glucose metabolic network. A higher hazard ratio (poor outcome) was observed in gene products that catalyze reactions in Gln catabolism and higher expression of glycolytic genes culminated in better patient survival (low hazard ratio).

Data information: Data in (A–D) are expressed as mean ± SEM, *$P < 0.05$, **$P < 0.01$, ***$P < 0.001$. In (D), 1-way ANOVA with Tukey's test was used to compare between cell lines, using OVCAR3 as the control.

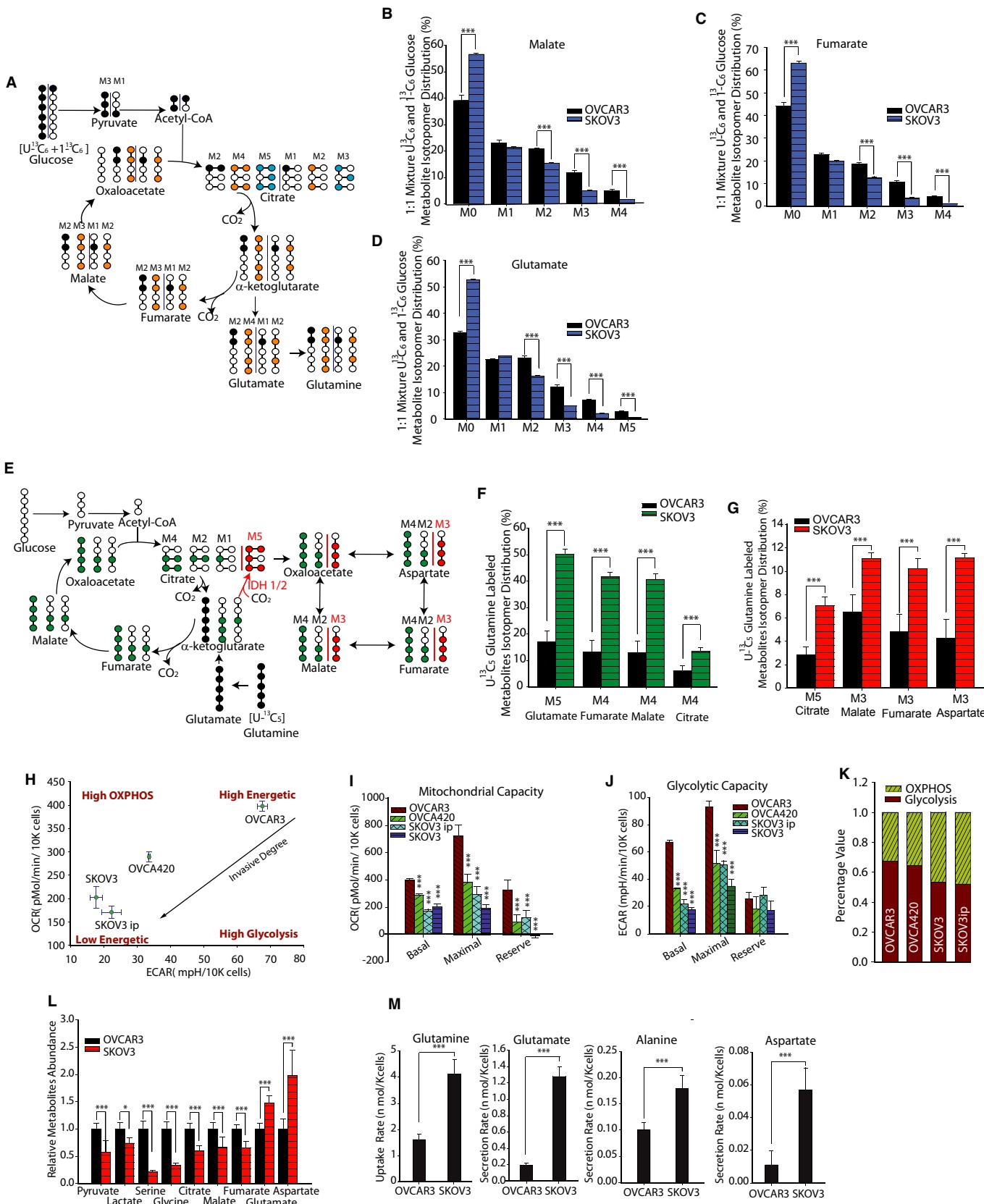

**Figure 2**

oxidative phosphorylation (OXPHOS, indicated by OCR) and higher levels of basal glycolysis (indicated by ECAR) than high-invasive OVCA cells (Fig 2H–J). Although most cancer cells exhibit high levels of glycolysis and functional mitochondrial oxidative phosphorylation, cancer cells from many tumors have dysfunctional mitochondria and increased glycolytic capacity. Using oligomycin, the protonophoric uncoupler FCCP, the electron transport inhibitor antimycin and the Glc analog 2-deoxyglucose (2-DG), we measured parameters indicative of basal, maximal and reserve mitochondrial and glycolytic capacities for OVCAR3, OVCA420, SKOV3 and SKOV3ip cells. Low-invasive OVCA cells had higher basal, maximal and reserve mitochondrial capacity than high-invasive OVCA cells (Fig 2I). However, reserve glycolytic capacity did not change with increasing invasiveness (Fig 2J). Essentially, low-invasive OVCA cells are in a higher energy-generation state than high-invasive OVCA cells. We then investigated the relative contribution of glycolysis and OXPHOS toward energy maintenance. To understand the cell invasiveness-sensitized energetic shift, we used oligomycin, an inhibitor of ATP synthetase that blocks the electron transport chain (ETC.), to force cancer cells to shift their energy-generation pathway from OXPHOS to glycolysis. The cellular ATP content did not reduce with addition of oligomycin (Supplementary Fig S3C and D). Therefore, the increased levels of glycolysis (indicated through lactate secretion) are likely a consequence of OXPHOS inhibition (Hao *et al*, 2010). From these data, we can conclude that the dominant pathway of energy-generation shifts from glycolysis to OXPHOS with increasing degrees of invasiveness (Fig 2K). Similar results were obtained by independently estimating the ratio between basal OCR and basal ECAR of different cancer cells using extracellular flux analysis. Indeed, the value of OCR/ECAR increases with increasing degrees of invasiveness (Supplementary Fig S3E).

To further verify differential metabolic rewiring of glycolysis and glutaminolysis in high- and low-invasive OVCA cells, we used GC-MS to quantify relative metabolite abundances in OVCAR3 and SKOV3 using targeted metabolomic analysis. Remarkably, low-invasive OVCA cells had higher quantities of glycolysis (pyruvate and lactate) and TCA cycle (citrate, malate and fumarate) metabolites than high-invasive OVCA cells (Fig 2L). This further confirms the increased OCR and ECAR observed in low-invasive cancer cells. Glutaminolysis levels, measured using metabolic isotope abundances of glutamate and aspartate, were markedly higher in high-invasive OVCA cells than in low-invasive OVCA cells (Fig 2L). Through secretomics of uptake/secretion of primary and secondary amino acids using ultra-high-performance liquid chromatography (UPLC), we found that high-invasive OVCA cells (SKOV3) had a higher Gln uptake flux, and a higher glutamate, alanine, aspartate, ornithine secretion flux through Gln supported anaplerosis pathway when compared to low-invasive OVCA cells (OVCAR3; Fig 2M, Supplementary Fig S3F and G). Notably, SKOV3 has a higher uptake flux of (i) cysteine, which may be used for glutathione generation, and (ii) serine, lysine and secondary amino acids (AABA) which may support cell proliferation. However, SKOV3 has a lower uptake rate of branched chain amino acids (BCAA; isoleucine and leucine); these metabolites can increase mitochondrial TCA cycle activity (Supplementary Fig S3F and G; Supplementary Table S1). Lower uptake of BCAA rules out BCAA catabolism as a part of the TCA cycle in high-invasive cells. The above results conclusively show that as invasiveness increases in OVCA cells, cells shift their dependence on Glc onto Gln, with respect to generation of TCA cycle metabolites.

## High-invasive cells are glutamine dependent for increased respiration, proliferation and redox balance

The above experiments show that high-invasive cancer cells exhibit higher utilization of Gln for maintenance of their TCA cycle metabolite pool. We further investigated whether Gln increases mitochondrial respiration in high-invasive OVCA cells. Adding Gln to Gln-deprived cells dramatically increases OCR in high-invasive OVCA cell lines (SKOV3 and SKOV3ip), whereas there was no observable change in the OCR of low-invasive OVCA cells (OVCAR3 and IGROV1) and only a marginal OCR increase in moderately invasive

---

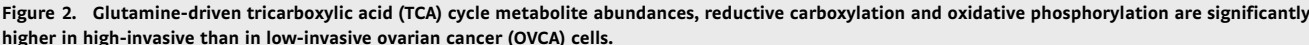

**Figure 2.  Glutamine-driven tricarboxylic acid (TCA) cycle metabolite abundances, reductive carboxylation and oxidative phosphorylation are significantly higher in high-invasive than in low-invasive ovarian cancer (OVCA) cells.**

A   Schematic of carbon atom transitions using 1:1 mixture of $^{13}C_6$ glucose and 1-$^{13}$C-labeled glucose. This allows estimations of the contribution of glucose toward TCA cycle metabolites and synthesis of glutamate and Gln. Black color represents labeled carbon on 1st TCA cycle, yellow color represents labeled carbon on 2nd TCA cycle, and blue color represents labeled carbon on 3rd TCA cycle.

B–D   Glucose's contribution to TCA metabolites and glutamate pool. Comparison of mass isotopomer distribution (MID) of malate (B), fumarate (C), glutamate (D) in high-invasive (SKOV3) and low-invasive (OVCAR3) cells cultured with a 1:1 mixture of $^{13}C_6$ glucose and 1-$^{13}$C-labeled glucose.

E   Schematic of carbon atom transitions using $^{13}C_5$-labeled Gln. Black color represents labeled carbon of Gln before its entry into TCA cycle. Green color represents Gln's direct effect on canonical TCA cycle, red color represents Gln's effect on TCA cycle through reductive carboxylation.

F   Gln's contribution to TCA cycle metabolites pool. Comparisons of MID of M5 glutamate, M4 fumarate, M4 malate, M4 citrate in OVCAR3 and SKOV3 cultured with $^{13}C_5$-labeled Gln.

G   Mass isotopomer analysis of isotopomers linked with reductive carboxylation. M5 citrate, M3 malate, M3 fumarate and M3 aspartate analysis indicates significantly higher reductive carboxylation fluxes in high-invasive than in low-invasive OVCA cells.

H   Basal oxygen consumption rate (OCR) and extracellular acidification rate (ECAR) were measured for OVCAR3, OVCA420, SKOV3 and SKOV3ip cell lines. Basal OCR is a measure of OXPHOS activity, and basal ECAR is a measure of glycolysis activity.

I   Using oligomycin, FCCP and antimycin, we estimated mitochondrial functional state in OVCA cells through maximal and reserve mitochondrial capacities.

J   Using 2DG, we estimated glycolytic functional state in OVCA cells through maximal and reserve glycolytic capacities.

K   Percentage of OXPHOS and glycolysis contribution to cancer cell's energetic demand.

L   Relative metabolite abundances were measured using GC-MS in OVCAR3 and SKOV3 cells.

M   Extracellular uptake/secretion fluxes of amino acids involved in glutaminolysis (Glutamine: Gln; Glutamate: Glu; Alanine: Ala; Aspartate: Asp) were measured using ultra-high-performance liquid chromatography.

Data information: Data are expressed as mean ± SEM, $n \geq 9$, *$P < 0.05$, **$P < 0.01$, ***$P < 0.001$. In (I) and (J), one-way ANOVA with Tukey's test was used to compare between cell lines, using OVCAR3 as the control.

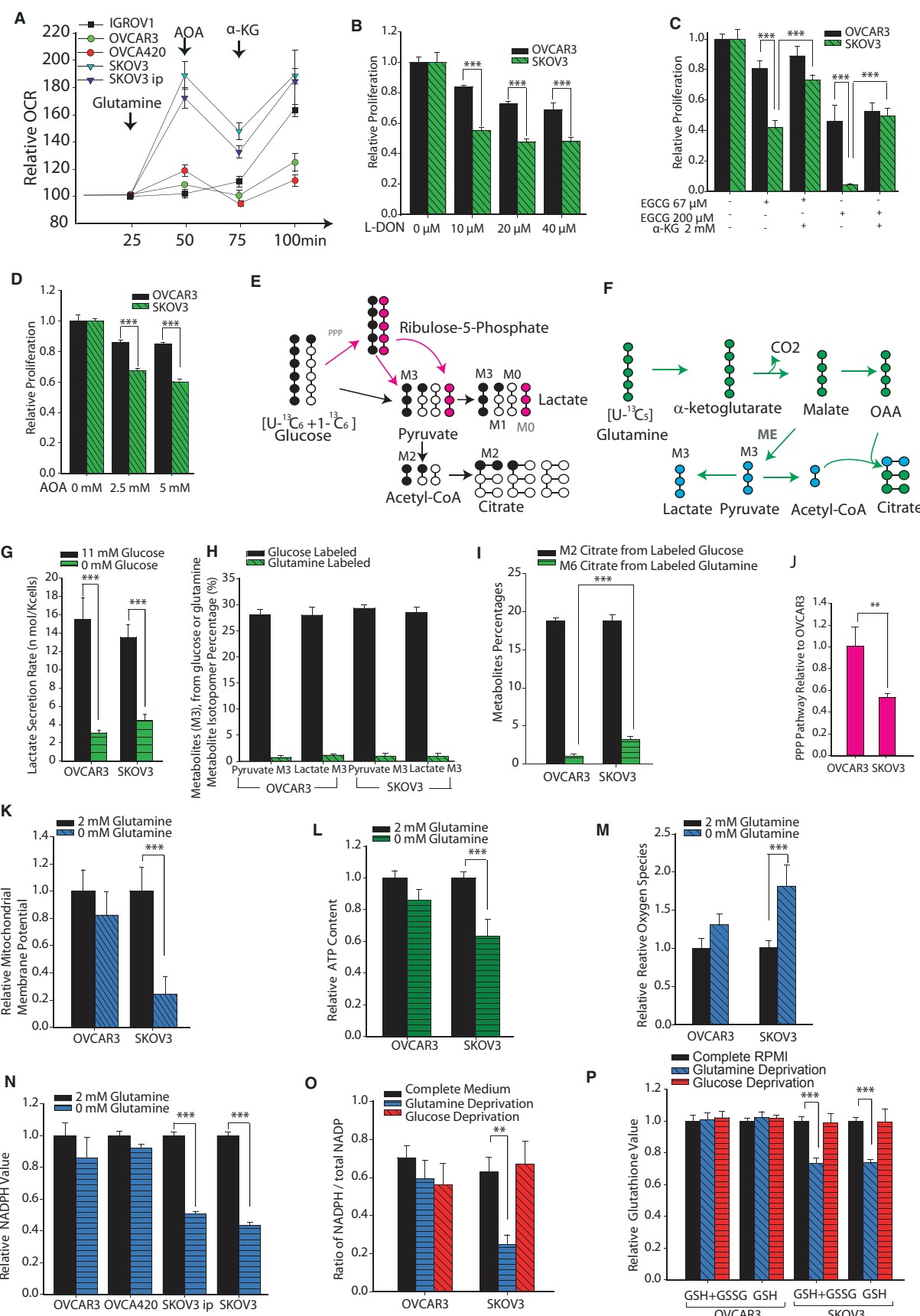

**Figure 3**

OVCA cells (OV420; Fig 3A). To establish whether the effects of Gln addition are due to its entry into the TCA cycle, we added amino-oxyacetate (AOA), an inhibitor of glutamate dependent amino-transferase, to block glutamate conversion to α-ketoglutarate (α-KG). The addition of AOA decreases OCR significantly in high-invasive cancer cells. Respiration is markedly rescued in high-invasive cancer cells by addition of dimethyl α-KG, a cell penetrant form of α-KG. In OVCAR3, IGROV1 and OVCA420, α-KG still increases OCR, since there is sufficient mitochondrial reserve capacity in these cell lines (Fig 3A).

Next, we determined whether high-invasive cancer cells have increased glutaminolysis levels to support proliferation. Using 6-diazo-5-oxo-L-norleucine (L-DON), a Gln analog which inhibits Gln conversion into glutamate, we found that high-invasive OVCA cells are highly sensitive to decreased glutamate, while only a marginal decrease in proliferation is observed in low-invasive OVCA cells (Fig 3B). To confirm the importance of Gln in supporting growth through the TCA cycle, we used AOA and epigallocatechin gallate (EGCG, an inhibitor of glutamate dehydrogenase) to inhibit glutamate's conversion into α-KG. A large decrease in the growth of high-invasive OVCA cell lines was observed following addition of AOA and EGCG (Fig 3C and D). These results suggest that high-invasive cell lines are significantly more Gln-dependent than low-invasive OVCA cells. To further substantiate that the supportive effect of Gln on cell growth occurs through glutaminolysis and entry of Gln into the TCA cycle, we added cell permeable dimethyl α-KG into the culture medium along with EGCG. Indeed, α-KG rescued the proliferation rate of high-invasive OVCA cells, whereas α-KG had no effect on the growth of low-invasive OVCA cells (Fig 3C). We confirmed that this rescue of proliferation is mediated through Gln

entry into the TCA cycle. The addition of ammonia and increasing concentration of other nonessential amino acids, including aspartate, asparagine, serine, glycine and alanine failed to rescue the high-invasive OVCA cells from Gln depletion (Supplementary Fig S4A).

We used 2-deoxyglucose (2-DG), a glycolysis inhibitor, to illustrate the effects of glycolysis on cell proliferation and metabolism. 2-DG reduced cell proliferation in both high-invasive (SKOV3) and low-invasive (OVCAR3) OVCA cells (Supplementary Fig S4B). Interestingly, adding Glc markedly decreased the OCR in low-invasive OVCA cells, demonstrating competition between glycolysis and oxidative phosphorylation for ATP generation (Supplementary Fig S4C). High-invasive cells showed a modest OCR decrease induced by Glc depletion. Consistent with previous results, these data confirm that Glc is a critical nutrient for low-invasive OVCA cells and that these cells are metabolically reprogrammed toward high Glc sensitivity (Supplementary Fig S4B and C).

It has been shown previously that a significant portion of Gln is diverted toward lactate secretion, via malate and malic enzyme (De-Berardinis *et al*, 2007). Because U-$^{13}$C$_6$ Glc produces M3 pyruvate and lactate through glycolysis (Fig 3E), whereas U-$^{13}$C$_5$ Gln produces M3 pyruvate and lactate through glutaminolysis and malic enzyme (Fig 3F), we analyzed $^{13}$C-labeled isotopologues of lactate and pyruvate using labeled Glc and Gln as tracers. The lactate secretion in Glc-depleted and complete media indicates that in both OVCAR3 and SKOV3 cancer cells, Glc is the major source of lactate (Fig 3G). There were significantly lower levels of M3 pyruvate, M3 lactate and M6 citrate, from U-$^{13}$C$_5$ Gln, than M3 pyruvate, M3 lactate and M2 citrate, from 1:1 mixture of U-$^{13}$C$_6$ Glc and 1-$^{13}$C-labeled Glc, for both cell lines (Fig 3H and I). These data suggest that lactate is mainly derived from Glc rather than from Gln in

**Figure 3. Glutamine has pleiotropic role of positively regulating respiration and maintaining redox balance selectively in high-invasive ovarian cancer (OVCA) cells.**

A    Oxygen consumption rate (OCR) was measured in high-invasive (SKOV3 and SKOV3ip), low-invasive (IGROV1 and OVCAR3) and moderately invasive (OVCA420) cells in media containing Gln, aminooxyacetate (AOA) and α-ketoglutarate (α-KG). OCR was normalized with value before injections.

B    Analysis of proliferation of OVCA cell lines with range of concentrations of 6-diazo-5-oxo-L-norleucine (L-DON), a Gln analog which inhibits Gln's conversion to glutamate. Data were normalized with complete media conditions without L-DON.

C    Analysis of proliferation of OVCA cell lines with range of concentrations of epigallocatechin gallate (EGCG, an inhibitor of glutamate dehydrogenase, which converts glutamate into α-KG) used to inhibit glutamate entering the tricarboxylic acid (TCA) cycle. α-KG was used to rescue reduction of cell proliferation by EGCG. Data were normalized with complete media conditions without EGCG.

D    Analysis of proliferation of OVCA cell lines when glutaminolysis was inhibited with range of concentrations of AOA. Data were normalized with complete media conditions without AOA.

E    Lactate and citrate generation from glucose through direct glycolysis or through both oxidative pentose phosphate pathway (PPP) and glycolysis. Black circles represent labeled carbons, empty circles represent unlabeled carbons, pink circles represent unlabeled carbons in oxidative PPP.

F    Lactate and citrate generation from glutaminolysis. Blue circles represent labeled carbons from malic enzymes, and green circles are labeled carbons from canonical TCA cycle.

G    Glucose deprivation effect on lactate secretion rate.

H    Comparison of M3 pyruvate and M3 lactate derived from either glucose (1:1 mixture of U-$^{13}$C$_6$ glucose and 1-$^{13}$C$_1$ glucose) or Gln (U-$^{13}$C$_5$ Gln) in high- and low-invasive OVCA cells. Since all conditions have complete media, their total pyruvate and lactate content should be the same.

I    Comparison of M2 citrate labeling from glucose and M6 citrate labeling from Gln in OVCAR3 and SKOV3 cells. Total pyruvate and lactate content in all conditions should be equal as explained for (H).

J    Oxidative pentose phosphate pathway fluxes estimated using the mathematical relation (percentage of unlabeled M0 lactate—percentage of labeled M1 lactate) to represent relative oxidative PPP fluxes in OVCAR3 and SKOV3 cells cultured with 1:1 mixture of $^{13}$C$_6$ glucose and 1-$^{13}$C-labeled glucose.

K    Role of Gln in maintaining mitochondrial membrane potential (MMP) levels in low- and high-invasive cells.

L    Gln maintains ATP content selectively in high-invasive OVCA cells.

M    Gln reduces reactive oxygen species induced by H$_2$O$_2$ in high-invasive cells, but not in low-invasive cells.

N    Gln maintains NADPH level in high-invasive OVCA cells.

O    Gln increases ratio of NADPH/total NADPH ratio selectively in high-invasive OVCA cells. Glucose is unable to provide enough reducing equivalents in high-invasive OVCA cells.

P    Gln/glucose deprivation's effect on total glutathione and reduced glutathione level in cancer cells.

Data information: Data in (A–N) are expressed as mean ± SEM, $n \geq 9$, **$P < 0.01$, ***$P < 0.001$. Data in (O, P) ($n \geq 6$) are expressed as mean ± SEM, *$P < 0.05$, **$P < 0.01$, ***$P < 0.001$.

OVCA cells. Interestingly, increased M6 citrate from labeled Gln indicates that malic enzyme is important for NADPH generation in high-invasive OVCA cells (Fig 3F and I). In contrast, low-invasive cancer cells had increased oxidative pentose phosphate pathway flux (PPP) compared to high-invasive cancer cells (Fig 3J), which was estimated using M0 and M1 labeling of lactate in cells cultured with 1:1 mixture of U-$^{13}C_6$ Glc and 1-$^{13}$C-labeled Glc. 1-$^{13}$C-labeled Glc loses the labeled carbon when it is routed through oxidative PPP and then is converted into M0 pyruvate (Gaglio *et al*, 2011). These results support the observation that Glc is critical in increasing PPP flux for generating NADPH in low-invasive cells.

Previous studies have suggested that c-myc and k-ras can induce oncogenic transformations and increase Gln utilization in tumors (Wise *et al*, 2008; Gaglio *et al*, 2011). However, c-myc protein expression levels have already been shown to be similar in both high-invasive (SKOV3) and low-invasive cell lines (OVCAR3), and no Kras mutations were detected in previous reports (Russo *et al*, 2003). Consistent with published data, we did not observe significant changes in the expression levels of c-myc, and k-ras in OVCAR3 and SKOV3 cell lines (Supplementary Fig S4D and E). Thus, we can conclude that the increased glutaminolysis observed in high-invasive OVCA cells is not due to increased expression of oncogenes c-myc and k-ras (Supplementary Fig S4D and E). Our data suggest that the increased glutaminolysis is associated with metabolic reprogramming induced by an increase in cancer cell invasiveness.

The above data demonstrate that Gln facilitates mitochondrial function in high-invasive OVCA cells. We then proceed to determine whether Gln is involved in maintaining mitochondrial membrane potential (MMP) and whether it increases antioxidant activity via glutathione under oxidative stress. Using tetramethylrhodamine methyl ester (TMRM), an MMP sensitive fluorescent dye, we measured MMP after adding $H_2O_2$ to the culture medium for ROS generation. Indeed, in high-invasive OVCA cells (SKOV3), MMP decreased significantly under Gln deprivation conditions. In contrast, there was no change in the MMP of low-invasive OVCA cells (OVCAR3) during Gln deprivation (Fig 3K), suggesting that Gln affects MMP only in high-invasive cancer cells. In agreement with previous results, Gln addition doubles the ATP content in high-invasive cancer cells, but shows only a marginal effect in low-invasive OVCA cells (Fig 3L). Gln addition dramatically reduces ROS induced by $H_2O_2$, in high-invasive cancer cells (SKOV3). However, in low-invasive cancer cells, Gln does not reduce ROS levels (Fig 3M). To further distinguish the differential role of Gln with respect to cancer invasiveness, we studied the role of Gln in NADPH and reduced glutathione (GSH) generation. NADPH is critical for the synthesis of GSH, an important antioxidant, by glutathione reductase. We observed that Gln deprivation decreases NADPH levels in high-invasive OVCA cells (SKOV3 and SKOV3ip), but not in low-invasive OVCA cells (OVCAR3; Fig 3N). Thus, Gln serves as an important source for NADPH, which decreases ROS, in high-invasive OVCA cells. Furthermore, the ratio of NADPH to total NADP (NADPH/total NADP), an indicator of the availability of reducing equivalents (NADPH), was markedly sensitive to Gln deprivation in high-invasive cells, but not in low-invasive OVCA cells (Fig 3O). To confirm Gln's role in scavenging ROS, we measured the levels of glutathione. Gln deprivation decreases total glutathione and GSH level in SKOV3 (Fig 3P). Interestingly, Glc

deprivation does not have a significant effect on MMP, ATP generation or ROS in high-invasive OVCA cells (Supplementary Fig S4F–I).

## Ratio of glutamine catabolism over anabolism correlates with invasiveness

To reveal the genetic cause behind dependence on Gln, we first analyzed the relative expression of Gln pathway genes in OVCA cell lines using KyotoOv38, an OVCA line gene expression dataset. Interestingly, glutaminase (GLS1, which converts Gln into glutamate) expression is approximately fourfold higher in high-invasive OVCA cells (SKOV3) compared to low-invasive OVCA cells (OVCAR3; Fig 4A). Moreover, low-invasive OVCA cells (OVCAR3) have a higher expression of Gln synthetase (GLUL) which catalyzes condensation of glutamate and ammonia into Gln (Fig 4A). The Western blot results confirm that high-invasive OVCA cells have relatively higher expression level of GLS1, but lower expression level of GLUL (Fig 4B). This metabolic reprogramming results in the Gln independence of low-invasive cancer cells. To further validate our findings that metabolic shifts in Gln utilization regulate cancer progression, we analyzed gene expression profiling and clinical outcome data from published studies. A correlation was discovered between progression-free survival of OVCA patients and the ratio of expression levels of Gln catabolism-related genes to those of Gln anabolism-related genes. Based on the noticeable effects of glutaminase expression levels on the prognosis of cancer patients, we found that competing activities of Gln catabolism and Gln anabolism act as potential biomarkers for predicting patient survival. The gene expression ratios GLS1/GLUL, GLUD1/GLUL and combined ratio (GLS1 + GLUD1)/GLUL are considered as valid representations of the Gln catabolic versus anabolic activities. Interestingly, patients above the 75th percentile of GLS1/GLUL and GLUD1/GLUL ratios had remarkably poorer prognoses than patients below the 25th percentile (Fig 4C–E). Therefore, the prevalence of Gln catabolism over Gln anabolism ((GLS1 + GLUD1)/GLUL) in OVCA patients has a conspicuous influence on their prognosis.

We further explored the reasons behind Gln independence in low-invasive OVCA cells, by converting 'Gln-independent' cells to a 'Gln-dependent' phenotype. Through investigation of cellular metabolism, energetics and invasiveness, we have demonstrated that low-invasive OVCA cells are Gln independent and high-invasive cells are Gln dependent. Our isotope labeling data reveal that low-invasive OVCA cells (OVCAR3) produce glutamate from other nutrient sources especially Glc (Fig 2D). Using labeled Glc, the glutamate isotope distribution data illustrate that low-invasive OVCAR3 cells have a significantly higher percentage of labeled (M2, M3, M4 and M5) glutamate than high-invasive OVCA cells (SKOV3; Fig 2D). These data suggest that although low-invasive OVCA cells are Gln independent, they use Glc for glutamate production to compensate for the lack of Gln.

We then performed nutrient-sensitized energetic shift experiments to confirm our premise that shifting a cell's metabolic focus from glycolysis to mitochondrial respiration will make low-invasive OVCA cells Gln dependent (Gohil *et al*, 2010; Weinberg *et al*, 2010). To convert Gln-independent cancer cells into Gln-dependent cells, we replaced 11 mM Glc in the culture medium with 11 mM galactose. Galactose enters glycolysis through the Leloir pathway, bypassing the hexokinase step, and is converted

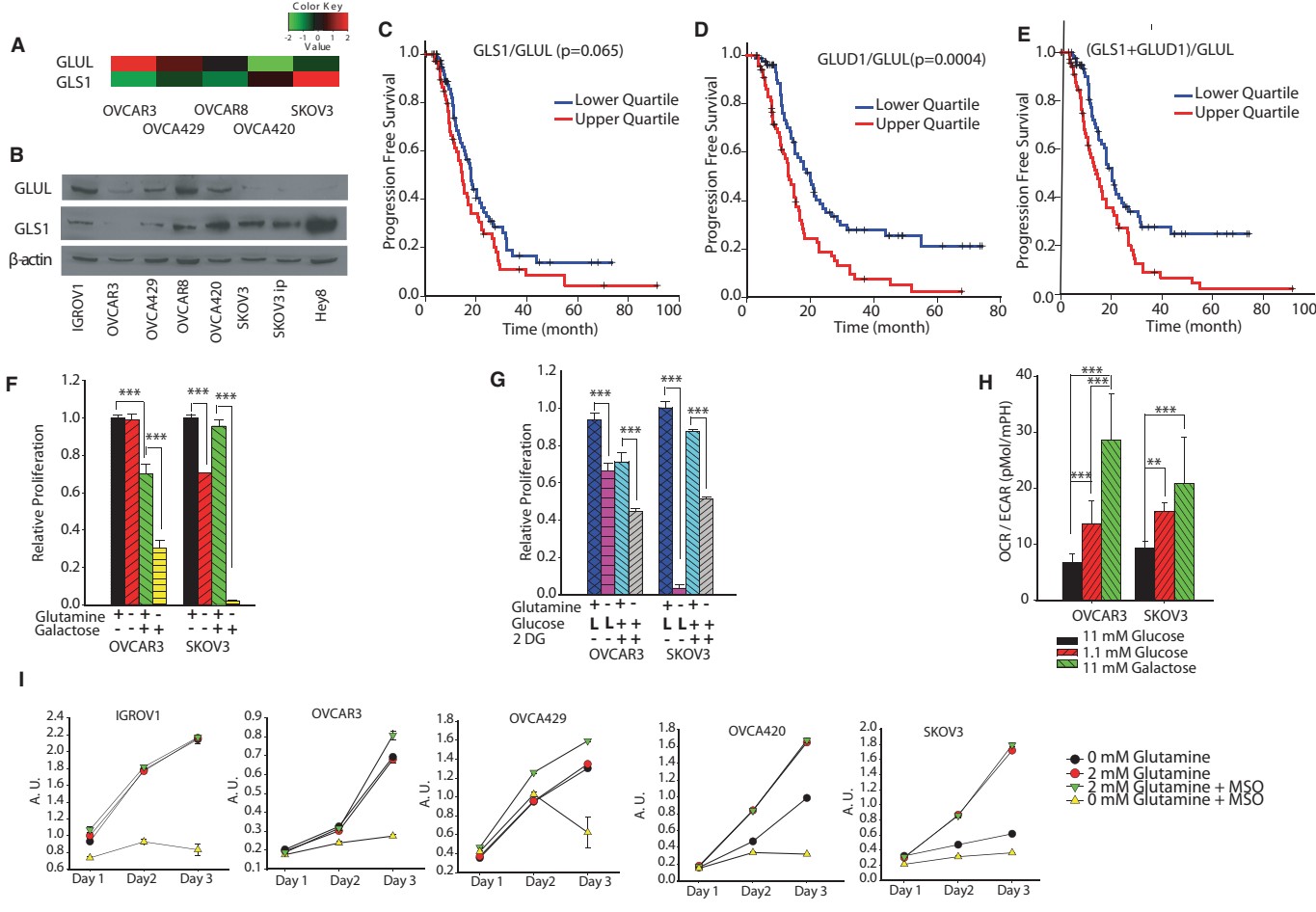

**Figure 4. Regulating ratio of glutamine catabolism over anabolism reduces tumorigenicity.**

A   Gene expression levels of Gln synthetase (GLUL) and glutaminase (GLS1) were determined using KyotoOv38, a database for gene expression of ovarian cancer cell lines.

B   Western blot of the GLUL and GLS1 protein expression levels for OVCA cell lines.

C–E   Progression free survival rate for OVCA patients, categorized according to GLS1/GLUL (C), GLUD1/GLUL (D), (GLS1 + GLUD1)/GLUL (E). The survival analysis is executed from comparison of upper quartile and lower quartile patients. (total patients *n* = 539).

F, G   Converting Gln-independent OVCA cells into Gln dependent by replacing glucose with galactose (F) or low glucose concentrations (L glucose) and 2-DG (10 mM) (G).

H   Galactose and low glucose's effect on oxygen consumption rate (OCR)/extracellular acidification rate.

I   Combined approach of targeting high-invasive OVCA cells by blocking glutamine's entry into the TCA cycle along with targeting low-invasive OVCA cells by inhibiting glutamine synthesis leads to pronounced reduction of cell growth. Targeting GLUL using MSO under Gln deprivation decreases OVCA cell growth. Arbitrary Units (AU) is proportional to cell number (*n* ≥ 10).

into glucose-6-phosphate at a very low rate compared to Glc (Weinberg *et al*, 2010). Hence, cells shift their energy metabolism toward mitochondrial glutaminolysis in the presence of galactose to compensate for reduced glycolysis (Gohil *et al*, 2010). We used galactose to assess whether shifting cellular energy metabolism toward the mitochondrial TCA cycle will give 'Gln-independent' cells a 'Gln-dependent' phenotype. The growth rate of low-invasive OVCA cells decreases when Glc is replaced with galactose in the presence of Gln (Fig 4F). Cell proliferation further decreases when Gln is depleted from the media, confirming the Gln-dependent phenotype of low-invasive cells when in the presence of galactose. Gln deprivation in high-invasive cells (SKOV3), in the presence of galactose, results in more than 90% cell death, further confirming the strong dependence of these cells on Gln (Fig 4F).

As an alternative strategy for converting 'Gln-independent' cells to a 'Gln-dependent' phenotype, we reduced glycolysis levels by decreasing either Glc concentration (reducing 11 mM Glc to 1.1 mM Glc) or using 2-DG (inhibiting glycolysis). Both low Glc concentration and 2DG converted low-invasive OVCA cells to Gln-dependent phenotype (Fig 4G). To determine the reason behind this increase in Gln importance in the presence of galactose or low Glc levels, we measured OCR and ECAR. Galactose and low Glc medium shifts the primary energy pathway of OVCAR3 and SKOV3 from glycolysis to OXPHOS, as evident from the increased OCR/ECAR. The increased OCR/ECAR enhances the function of Gln in the TCA cycle (Fig 4H; Supplementary Fig S5A and B). We further reasoned that Glc is used for Gln synthesis in low-invasive cancer cells, causing their Gln-independent phenotype. Thus, these cells use a *de novo* synthesis mechanism for Gln synthesis and are not sensitive to Gln

withdrawal. Indeed, when both Gln uptake and Gln synthesis pathways are inhibited, Gln pools in both high- and low-invasive OVCA cells are depleted, resulting in significant cell growth reduction for all OVCA cells (Fig 4I).

## Clinical significance of glutamine catabolism and therapeutic effectiveness of GLS1 siRNA in ovarian cancer models

To further evaluate the clinical significance of the ratio of Gln catabolism over anabolism, we assessed GLS1 and GLUL protein expression in epithelial OVCA samples ($n = 139$) using tissue microarrays (TMA) and correlated it to survival outcome. High GLS1 protein expression was associated with worse overall survival, whereas high GLUL protein expression was associated with better survival (Fig 5A). In concordance with our previous results obtained from gene expression data from TCGA, we found that the ratio of Gln catabolism over anabolism (GLS1/GLUL) through protein expression was also significantly related to worse overall survival ($P < 0.001$).

Next, we examined the biological effects of GLS1 siRNA in Gln-independent (IGROV1) and Gln-dependent (SKOV3ip1) cells in mouse models. Before initiating *in vivo* experiments, we checked the efficacy of siRNA for silencing GLS1 in SKOV3ip1 cells using targeted GLS1 siRNA. Transfection of SKOV3ip1 cells with the targeted siRNA resulted in >70% decrease in GLS1 mRNA levels (Fig 5B). Trypan blue dye assay confirmed that transfection of cells with siRNAs did not affect cell viability after 48 h of transfection, suggesting that siRNA was not toxic to cells. SiRNA was incorporated into the neutral nanoliposome DOPC (1,2-dioleoyl-sn-glycero-3-phosphatidylcholine); for therapy experiments, siRNA administration was started 1 week after tumor cell injection subcutaneously. Mice were divided into the following two groups ($n = 5$ mice per group): a) control siRNA-DOPC and b) GLS1 siRNA-DOPC. All of the animals were euthanized 2 weeks after inoculation. As seen in Fig 5C–E, treatment with GLS1 siRNA-DOPC in Gln-independent IGROV1 tumor-bearing mice did not affect tumor volume or deep tumor infiltration through the muscle layers compared to control siRNA treatment. In contrast, GLS1 gene silencing in Gln-dependent SKOV3ip1 tumor-bearing animals showed significant reduction in both tumor weight (60%; $P = 0.007$) and tumor volume (75%; $P = 0.008$) when compared to control siRNA-DOPC treated groups. Moreover, GLS1 siRNA-DOPC blocked tumor infiltration into surrounding tissues (Fig 5F–H).

## Glutamine regulates cancer cell invasiveness through STAT3 activity in high-invasive ovarian cancer cells

To explore the underlying mechanisms of Gln-induced increased invasiveness, we first assessed the effect of Gln on the invasive capacity of OVCA cells. Consistent with our working model showing the selective role of Gln in high-invasive cancer cells, we found that Gln directly maintains cancer cell's invasiveness. Gln significantly increases the invasive and migratory capacity of SKOV3, but not of OVCAR3 (Fig 6A, Supplementary Fig S6A). On the other hand, glucose deprivation only has an effect on migratory capacity (Supplementary Fig S6A and B). Interestingly, inhibiting the entry of Gln into the TCA cycle with LDON, AOA, EGCG or Bis-2-(5-phenylacetamido-1,3,4-thiadiazol-2-yl) ethyl sulfide (BPTES, GLS 1

inhibitor) significantly reduces the invasive capacity of the high-invasive SKOV3 cells (Fig 6A). In line with these findings, the invasive capacity of SKOV3 was restored upon addition of α-KG under Gln depletion conditions. Similar results were obtained upon addition of α-KG when Gln entry into the TCA cycle was blocked by EGCG. Using rotenone [inhibits complex I of mitochondrial electron transport chain (ETC.)], we confirmed that reducing TCA cycle activity indeed decreases the invasiveness of OVCA cells. Collectively, these results demonstrate that the pleiotropic role of Gln in tumor growth, invasion and survival within high-invasive OVCA cells is dependent on glutaminolysis and entry of Gln into the TCA cycle.

To further test the hypothesis that Gln mediates oncogenic transformations in high-invasive cells by regulating growth factor pathways, we assessed the levels of phospho-proteins in key signaling pathways (Fig 6B, Supplementary Fig S6G). Confirmation that glutaminolysis has a differential effect on growth factor pathways of high-invasive OVCA cells would further support our premise that Gln maintains invasiveness in OVCA cells by regulating these pathways. During Gln starvation conditions, we observed that the greatest differential effect between high- and low-invasive cells occurs in the STAT3 pathway (Fig 6B). Signal transducer and activator of transcription 3 (STAT3) is a latent cytoplasmic transcription factor activated in response to growth factors and cytokines (Aggarwal *et al*, 2009; Yue *et al*, 2012). STAT3 is highly expressed in OVCA and has been shown to confer drug resistance and induce growth-promoting effects and is implicated in the malignant transformation of cancer cells (Burke *et al*, 2001; Duan *et al*, 2006; Gest *et al*, 2012). The tyrosine phosphorylation of STAT3 through JAK (Janus kinases), RTK (receptor tyrosine kinases) and non-RTK activates STAT3, which then translocates to the nucleus and promotes gene expression to mediate oncogenic transformations, including aerobic glycolysis (Demaria *et al*, 2010). In contrast, the serine phosphorylation of STAT3 has been found to promote oxidative phosphorylation in mitochondria through interaction with mitochondrial complexes (Wegrzyn *et al*, 2009; Phillips *et al*, 2010). We observe that in complete medium, STAT3 phosphorylation at tyrosine 705 (Y705) is elevated in the high-invasive cell lines (Fig 6B). Gln deprivation significantly decreases the levels of Y705 in the high-invasive OVCA cells, but has no effect in low-invasive OVCA cells (Fig 6B, Supplementary Fig S6C), while Glc deprivation reduces Y705 levels in the intermediately invasive cells as well as the high-invasive cell lines (Supplementary Fig S6C). The level of Y705 parallels the abundance of Jak1 and phospho-Src in similar conditions (Fig 6B). EGFR, JAK tyrosine kinase, Src tyrosine kinase and Erk1/2 play a dominant role in STAT3 phosphorylation and are linked with cancer cell proliferation, metabolic regulation and metastasis (Chung *et al*, 1997; Garcia *et al*, 2001; Silva, 2004; Yue *et al*, 2012; Gough *et al*, 2013). As the degree of invasiveness increases, phosphorylation of EGFR and Erk 1/2 also increases when in complete medium (Fig 6B). Glc deprivation has no effect upon STAT3 phosphorylation of serine 727 (S727) in any of the cells lines, whereas Gln deprivation results in a significant decrease in S727 level in the high-invasive cell lines (Fig 6B, Supplementary Fig S6D). The deprivation of other amino acids, such as serine, glycine, asparagines and aspartate, did not affect the STAT3 phosphorylation in OVCA cells (Supplementary Fig S6F). Since Gln depletion decreases STAT3 phosphorylation, we inquired whether overexpression of STAT3 could attenuate the Gln deprivation-induced reduction in proliferation. Significantly, when STAT3

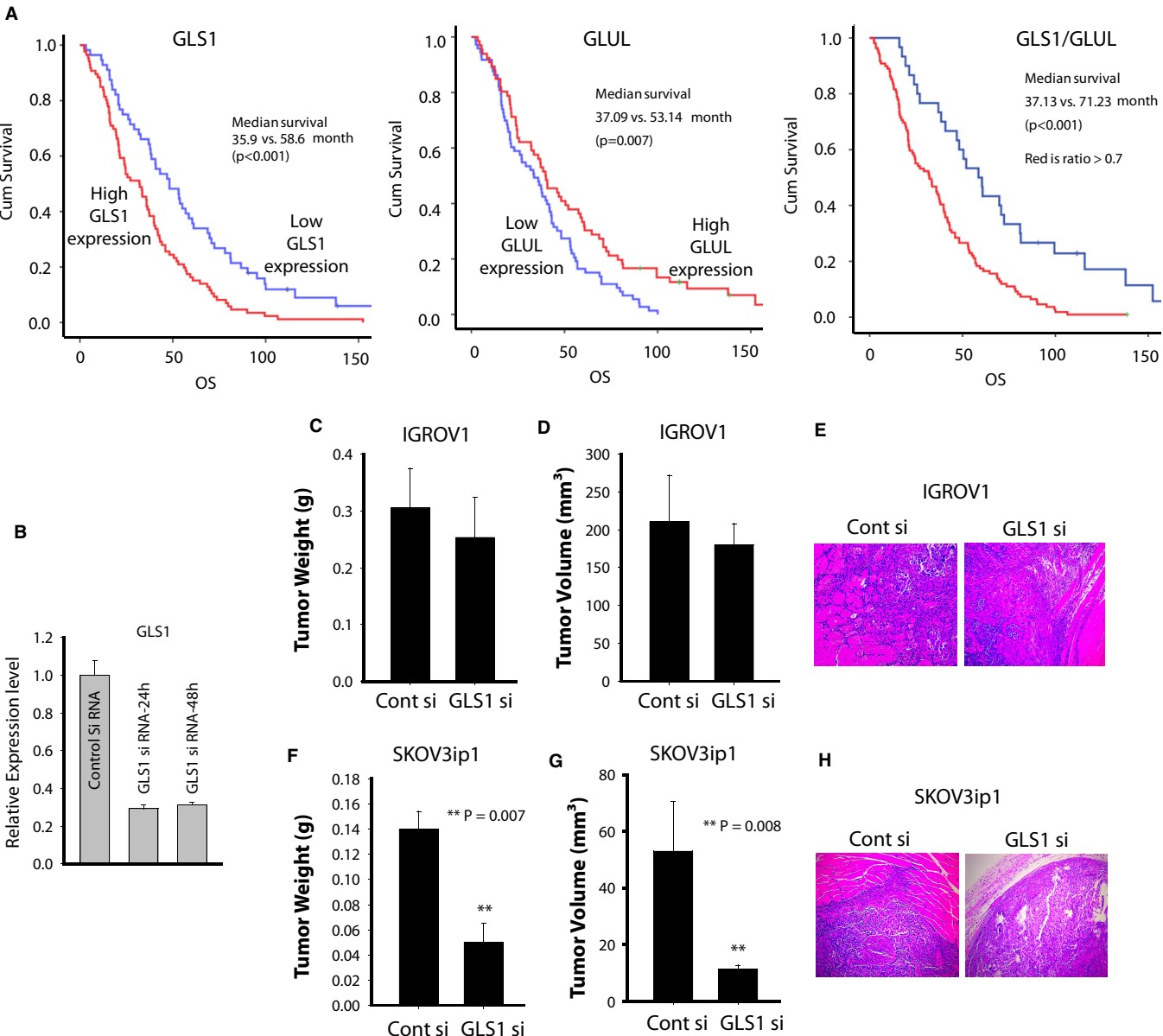

**Figure 5. Glutaminase activity is required for tumor growth and invasion in high-invasive ovarian cancers (OVCAs).**

A    Kaplan–Meier curves of disease-specific survival for patients with epithelial OVCA (*n* = 139) based on GLS1 and Gln synthetase protein expression. The log-rank test (two sided) was used to compare difference between groups.

B    Following transfection with either GLS1 siRNA or control siRNA, mRNA levels were assessed with qRT-PCR.

C–H  Therapeutic efficacy of siRNA-mediated GLS downregulation: (C) tumor nodule (D) volume (E) pattern of invasion of low-invasive OVCA cell line (IGROV1); (F) tumor nodule (G) volume and (H) pattern of invasion of SKOV3ip1 cells in nude mouse models. Following subcutaneous injection of nude mice with $2.0 \times 10^6$ IGROV1 or SKOV3ip1 cells, mice were randomly allocated to one of the following groups: control siRNA-DOPC or GLS1 siRNA-DOPC. Treatment was started 3 days after tumor cell injection, and siRNA-liposomes were administered twice weekly at a dose of 150 μg/kg body weight and continued for 2 weeks. At the time of sacrifice, mouse weight, tumor weight and tumor volume were recorded. Statistical analysis for tumor weights was performed by Student's *t*-test. **$P$ = 0.007 and **$P$ = 0.008 compared with control siRNA-DOPC.

is overexpressed in SKOV3 cells through transfecting with a constitutively active form of STAT3 (stat3c), we found that it rescues high-invasive OVCA cells (SKOV3) from Gln deprivation-induced reduction of cellular growth (Fig 6C). Importantly, this confirms our premise that Gln exerts oncogenic transformations in high-invasive cancer cells by differentially activating STAT3 in these cells compared to their low-invasive counterparts. Strikingly, in line with

our previous findings in which we found that Gln entry into the TCA cycle maintains invasiveness, we found that levels of Y705 are maintained by α-KG and oxaloacetate (OAA) under Gln deprivation conditions (Fig 6D and E). Inhibiting TCA cycle activity through prevention of Gln entry into the TCA cycle, using BPTES or through rotenone inhibition of the ETC., results in a decrease of Y705 levels (Fig 6F). This further implicates glutaminolysis in maintaining

growth factor signaling. We found that OAA and α-KG fail to rescue S727 levels. Similarly, the addition of BPTES and rotenone produced no change in S727 levels, indicating that serine S727 levels are not dependent on TCA cycle activity (Fig 6D–F).

AG490, a Jak inhibitor, is known to modulate the phosphorylation level of STAT3. Notably, we found that AG490 substantially decreases the invasive capacity of high-invasive cancer cells

(Fig 6G) but has a marginal effect on low-invasive cells. Furthermore, in high-invasive OVCA cells, Gln/Glc deprivation results in increased phosphorylation levels of p38 MAPK, a protein kinase central to regulation of MAPK signaling pathway cascades; increased phosphorylation level of ACC, an enzyme integral to lipid metabolism regulated through AMPK signaling pathway, as well as decreased FAK phosphorylation levels for cellular adhesion.

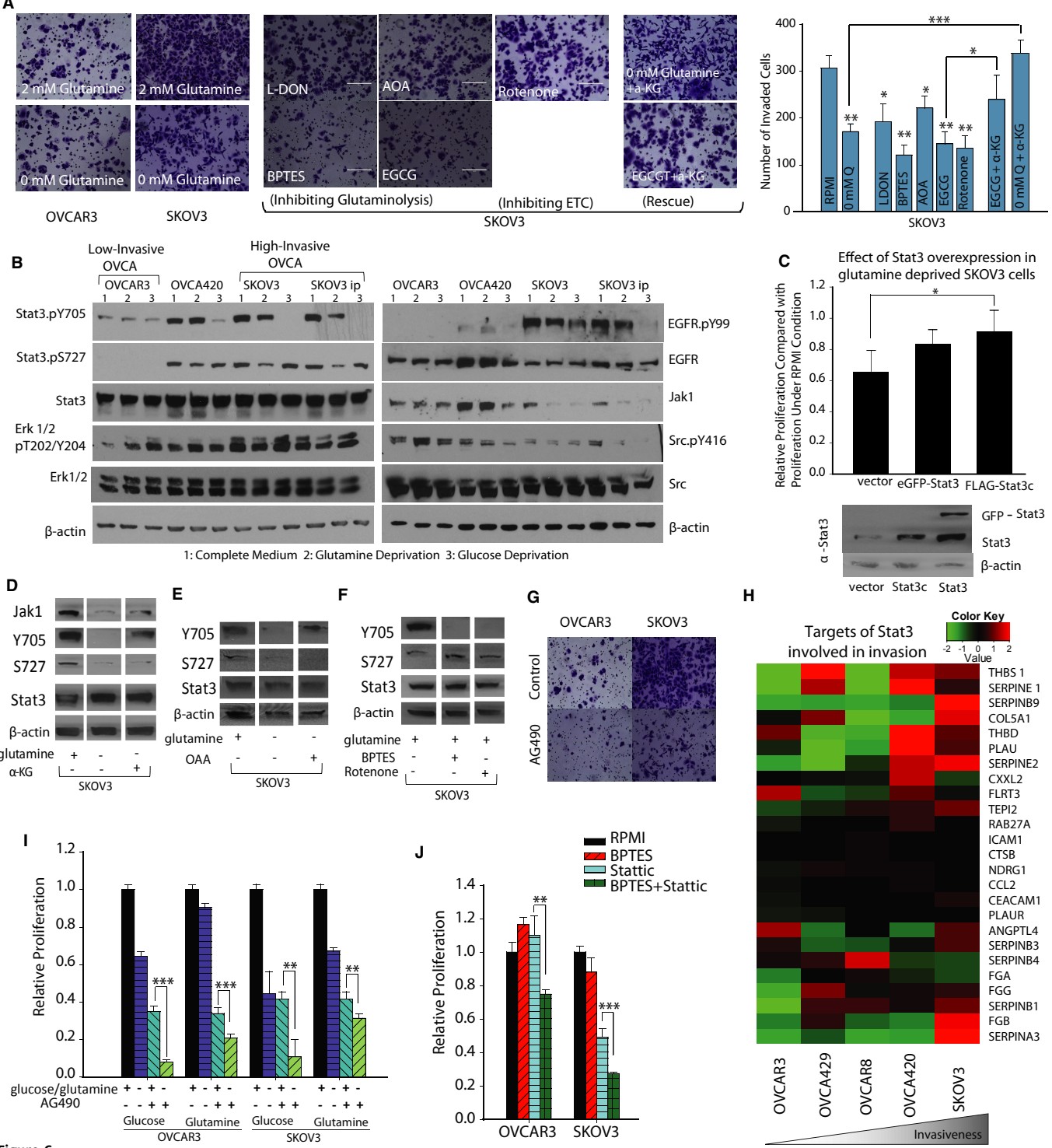

**Figure 6**

Additionally, Glc/Gln deprivation did not have significant effect on TSC2/GSK3 signaling, which regulates energy metabolism. However, Glc, but not Gln deprivation enhanced autophagy in all cell lines indicated by increased LC 3β-II and AKT phosphorylation, which regulate Glc metabolism in cancer cells (Supplementary Fig S6G). We further analyzed the relative expression of STAT3 target genes in OVCA cell lines using the KyotoOv38 dataset. Consistent with our observations, STAT3 is found to be important for OVCA invasion since it targets the expression of invasion-related genes, including THBS1, SERPINE1/2, COL5A1, THBD and PLAU (Fig 6H). Indeed, STAT3 inhibition decreases cell proliferation and enhances cell death within low- and high-invasive OVCA cells in both Glc and Gln deprivation conditions (Fig 6I). A combination of STAT3 inhibition and nutrient deprivation (Glc or Gln) has a stronger effect due to the pleiotropic roles of both STAT3 and nutrients. In line with these findings, inhibiting both STAT3 activity (through addition of stattic, a STAT3 inhibitor) and conversion of Gln into glutamate (through BPTES, a glutaminase inhibitor) results in a pronounced reduction of viability in high-invasive cancer cells (Fig 6J).

### Glutamine-dependent anaplerosis regulates glycolysis in high-invasive cancer cells through STAT3 phosphorylation

To expand our findings on the differential regulation of Gln metabolism in high-invasive cancer cells, we examined both the effect of Gln on glycolysis and the effect of Glc on glutaminolysis. We hypothesized that Gln's effects on the metabolic rewiring of high-invasive OVCA cells act through STAT3 phosphorylation. We first cultured OVCA cells in complete and Gln-depleted media to measure Glc uptake and lactate secretion levels. Notably, we observed competitive regulation of Glc by Gln in high-invasive cancer cells. Gln addition significantly downregulates glycolysis (Glc uptake and lactate secretion) in high-invasive cancer cells, but has no effect on low-invasive OVCA cells (Fig 7A and B). Similar conclusions were drawn from experiments on OVCA429 and OVCAR8 cells (Supplementary Fig S7A). To confirm whether the effect of Gln on glycolysis acts through its entry into the TCA cycle, we added AOA, an inhibitor of Gln entry into the TCA cycle. Addition of AOA selectively increases glycolysis in SKOV3, but not in OVCAR3 (Fig 7C and D). The addition of α-KG reverses this metabolic coupling in high-invasive cells (SKOV3). The reduced glycolysis observed in low-invasive cells (OVCAR3) upon α-KG addition is the result of increased TCA cycle activity (Fig 3A). We next cultured OVCA cells in Glc-depleted and complete media and used an UPLC to measure uptake/secretion fluxes of Gln, glutamate, aspartate and alanine. Glc deprivation increases Gln uptake in both high- and low-invasive OVCA cells. Conversely, only in high-invasive OVCA cells is there a significant increase in the secretion of glutamate, aspartate and proline, which can be derived from Gln (Fig 7E and F, Supplementary Fig S7B and C). Moreover, there is an increased uptake of cysteine (used for glutathione generation) and arginine (used in the urea cycle and protein synthesis) in high-invasive OVCA cells (Supplementary Fig S7D and E). For OVCAR3, Glc deprivation decreases alanine secretion due to decreased pyruvate generation (Fig 7E). Further, Glc deprivation increases serine uptake in both cell lines (Supplementary Fig S7D and E). This may be due to the absence of Glc-mediated serine synthesis for the SOG (serine, one-carbon cycle, glycine synthesis) pathway to generate NADPH, ATP and purines. Therefore, cancer cells will uptake more exogenous serine to meet their nutritional requirement for survival (Tedeschi *et al*, 2013; Maddocks *et al*, 2013).

To delineate STAT3-induced metabolic alterations from Gln driven metabolic pathways in high-invasive cancer cells, we used AG490 to measure glycolytic and OXPHOS activity in OVCA cells.

---

**Figure 6.  Glutamine mediates oncogenic transformations in high-invasive cells by regulating STAT3 activity.**

A   Comparison of OVCAR3 invasive capacity in Gln depleted and complete media conditions. Invasive capacity of SKOV3 is measured under complete media, Gln depleted media, and drugs inhibiting Gln's entry into tricarboxylic acid (TCA) cycle. L-DON, BPTES, EGCG, AOA, and Rotenone decrease SKOV3's Matrigel invasive capacity. α-Ketoglutarate (α-KG) addition under glutamine deprivation or EGCG conditions rescues SKOV3's invasive capacity. Data are expressed as mean ± SEM, *P < 0.05, **P < 0.01, ***P < 0.001, n ≥ 4.

B   Activation of Stat3 through tyrosine-705 phosphorylation (Stat3.pY705) is elevated in high-invasive ovarian cancer (OVCA) cells. The phosphorylation level of EGFR and Erk 1/2 increases with increasing degree of invasiveness. Tyrosine kinase signaling pathway activities can be affected by metabolic stress and in high-invasive cells that are glutamine and glucose deprived. Stat3.pY705 is reduced along with total Jak levels. Stat3.pY705 levels are only reduced in the OVCA420 cell lines upon glucose deprivation, while the regulation of Stat3 phosphorylation by metabolic stress is absent in the low-invasive cell line, OVCAR3. Stat3 serine-727 phosphorylation (Stat3.pS727) is also reduced along with phosphorylation level of Erk 1/2 in the high-invasive cell lines and only in response to glutamine deprivation. β-actin as loading control.

C   Cell proliferation in high-invasive OVCA cell line is reduced when glutamine starved. Proliferation can be partially rescued by overexpression of transgenic Stat3 and a constitutively active mutant of Stat3, Stat3c, mean ± SD, n = 3, *P < 0.05. β-actin as loading control.

D   α-KG addition rescues STAT3 tyrosine phosphorylation and Jak1, but cannot rescue Stat3 serine-727 phosphorylation. β-actin as loading control. Western blot figures are cut from same gel, but lanes are rearranged to show complete media conditions in first column, glutamine deprivation in second and α-KG addition in the third column. The full Western blot images of the gel and the detailed description are included in the source file.

E   OAA addition rescues STAT3 tyrosine phosphorylation. β-actin as loading control. Western blot figures are cut from same gel, but lanes are rearranged to include only the relevant conditions. The full Western blot images of the gel and the detailed description are included in the source file.

F   BPTES and rotenone inhibit STAT3 phosphorylation at tyrosine 705 (Y705). β-actin as loading control. Western blot figures are cut from same gel, but lanes are arranged to show only the relevant conditions. The full Western blot images of the gel and the detailed description are included in the source file.

G   Inhibition of Stat3 decreases OVCA metastasis. Treatment of SKOV3 cells with AG490 results in reduced invasion.

H   Gene expression levels of targets of STAT3 involved in invasion were determined using KyotoOv38 in five OVCA cells. High-invasive OVCA cells had higher gene expression of invasive genes.

I   The addition of AG490, a Stat3 inhibitor enhances the effect of glucose and glutamine deprivation on proliferation in OVCA cell lines.

J   The addition of stattic, inhibitor of STAT3, has a combinational cytotoxic effect on both cell lines.

Data information: Data in (I) and (J) are expressed as mean ± SEM, n ≥ 8, *P < 0.05, **P < 0.01, ***P < 0.001.
Source data are available online for this figure.

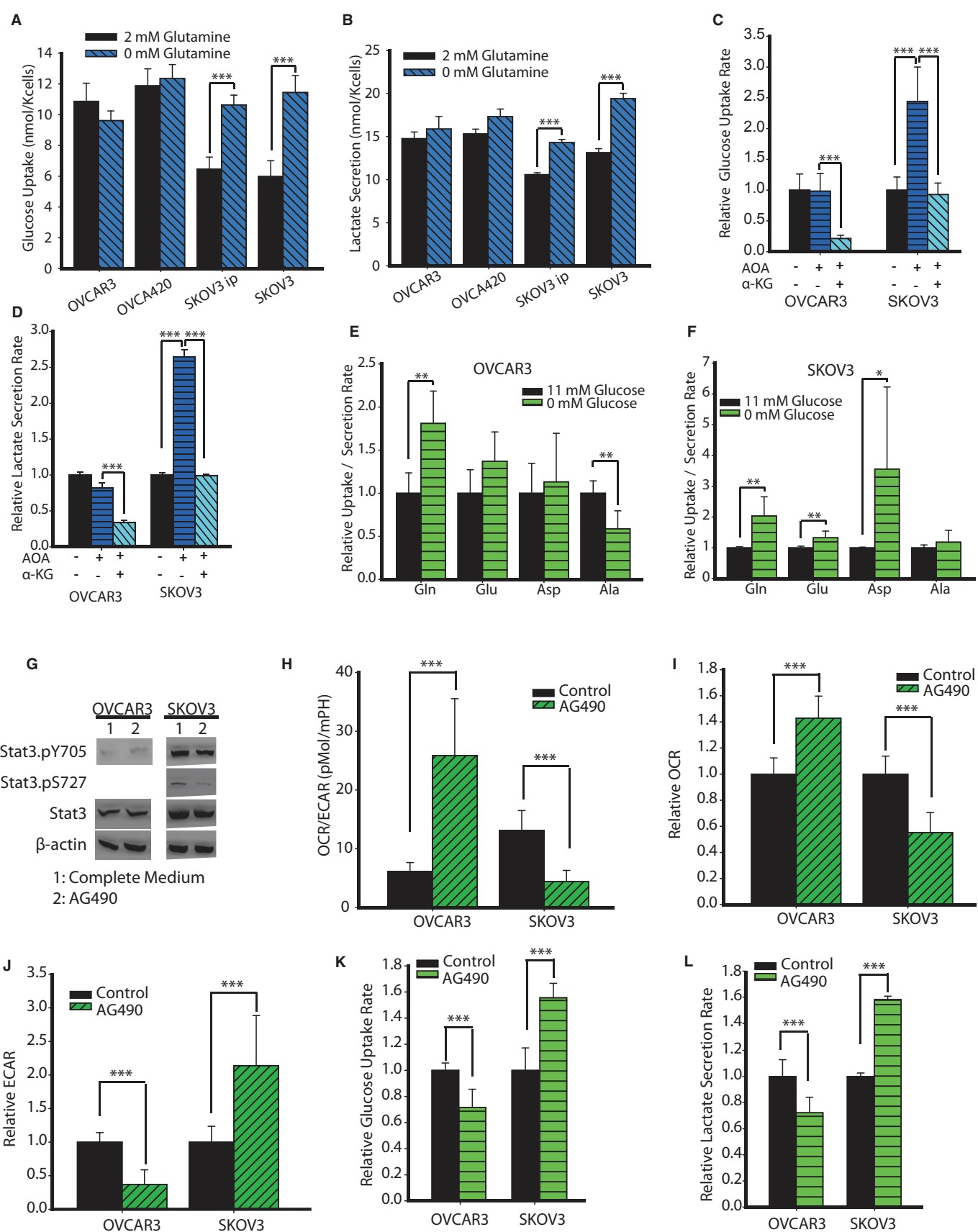

**Figure 7**

We found that AG490 decreases STAT3 serine phosphorylation in high-invasive OVCA cells (SKOV3) and marginally increases STAT3 tyrosine phosphorylation in low-invasive OVCA cells (OVCAR3; Fig 7G). Consistent with our hypothesis, inhibition of STAT3 phosphorylation decreases OCR in high-invasive OVCA cells (Fig 7H–J) along with a compensatory increase in glycolysis. Surprisingly, we found that AG490 decreases glycolysis in low-invasive OVCA cells, whereas it increases glycolysis in high-invasive OVCA cells (Fig 7K and L). Taken together, these results show that STAT3 interacts with the metabolic switch that shifts low-invasive OVCA cells from glycolysis to OXPHOS. Furthermore, AG490 induces metabolic reprogramming in high-invasive OVCA cells by decreasing STAT3 serine phosphorylation, thereby decreasing OXPHOS and increasing glycolysis.

## Discussion

The role of Gln in regulating cancers with varying degree of invasiveness has been poorly understood. Understanding the differential role of Gln with respect to increasing cancer invasiveness could lead to the development of a metabolic regulation strategy for targeting heterogeneous populations of cancer cells. Recent studies have indicated a differential metabolic wiring in high-invasive cancer cells compared to low-invasive cancer cells (Nomura *et al*, 2010). High-invasive cancer cells were shown to have increased expression of monoglycerol lipase, a protein which regulates the fatty acid network that initiates oncogenic signaling and promotes tumor pathogenicity (Nomura *et al*, 2010). For OVCA cells under anoikis conditions, we recently reported that pyruvate increases the migratory capacity of highly invasive OVCA cells' migration ability, thus implicating mitochondrial activity in the development cancer metastasis. Kim et al. showed that inhibiting mitochondrial function reduces cancer cells' invasive capacity (Kim & Wirtz, 2011, 2013). Here, we reveal that Gln is a key mitochondrial substrate for driving cancer metastasis. Our work defines the essential role of Gln in high-invasive OVCA cells and presents a viable therapeutic strategy to target OVCA tumors with variable degrees of invasiveness.

We show here for the first time that high-invasive OVCA cells are markedly dependent on Gln for cell proliferation, whereas low-invasive OVCA cells are Gln-independent for growth. As the degree of cancer invasiveness increases, nutrient dependence shifts from Glc to Gln (Fig 8). In both low and high-invasive OVCA cells, we found that Glc contributes more than 50% of pyruvate (M3 pyruvate and lactate was >25% in 1:1 mixture of $^{13}C_6$ Glc and 1-$^{13}$C-labeled Glc). On the other hand, in high-invasive OVCA cells, Gln contributes more than 50% of glutamate and 40% of TCA cycle fluxes, whereas in low-invasive OVCA cells, Gln contributes around 15% of glutamate and TCA cycle fluxes. Moreover, Gln addition increases the OCR in high-invasive cells (>80% in SKOV3 and SKOV3ip; Fig 3A). A highlight of our *in vitro* mechanistic studies and *in vivo* model is the finding that the inhibition of glutaminolysis is more detrimental to high-invasive OVCA cells than their low-invasive counterparts. GLS1-targeted siRNA significantly decreased cancer growth and invasion in mice bearing ovarian tumors derived from glutamine-dependent (high-invasive, SKOV3ip1) compared to glutamine-independent (low-invasive, IGROV1) cells. Our results substantiate the hypothesis that Gln is essential for anaplerosis in the TCA cycle and cell survival only in high-invasive cancer cells. In particular, Gln increases glutathione synthesis and reduces ROS differentially in high-invasive cells compared to low-invasive cells. Thus, high-invasive cells are dependent on Gln, which protects mitochondrial integrity by synthesizing glutathione to reduce ROS and promote cancer cell survival. Moreover, we found that Gln promotes cancer invasion in high-invasive cells. However, with increasing cancer invasiveness, there is a decrease in both glycolytic and basal mitochondrial capacity. This could be due to a shift in the role of nutrients (Glc and Gln) from energy generation to biosynthesis.

Herein, we provide previously unidentified evidence that Gln maintains invasive cancer phenotypes by regulating factors controlling the oncogenic transformations in cancer cells. STAT3 is involved in cellular differentiation, antiapoptotic response, metastasis and large-scale signaling system (Turkson & Jove, 2000; Levy & Lee, 2002; Schindler *et al*, 2007). The constitutive canonical tyrosine phosphorylation of STAT3 is required for transcriptional downstream activation of several oncogenes in the nucleus (Yu *et al*, 1995, 2009; Kataoka *et al*, 2008). We believe our results are the first to show that Gln deprivation regulates the phosphorylation of STAT3, an oncogenic transcription factor, and thereby induces rewiring of cancer metabolic pathways (Fig 8). As the degree of cancer invasiveness increases, the STAT3 phosphorylation cascade is altered. In low-invasive OVCA cells, Src and Jak1 increase STAT3 tyrosine phosphorylation. Whereas, EGFR, along with Src and Jak1, controls the tyrosine phosphorylation levels of STAT3 in high-invasive OVCA cells. Our results suggest that the tyrosine phosphorylation levels of STAT3 are dependent on Gln's entry into the TCA cycle, which further confirms our findings that glutaminolysis positively regulates invasiveness in OVCA cells. Furthermore, Erk1/2 is highly phosphorylated in high-invasive OVCA cells, resulting in promotion of STAT3 serine phosphorylation. Nutrients also play important roles in regulation of cell signaling pathways and rewire cancer metabolism (Fig 8). Deprivation of both Glc and Gln decreases

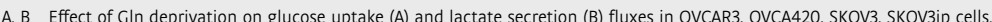

**Figure 7.  Glutamine's effect on metabolic rewiring in high-invasive ovarian cancer (OVCA) cells is through STAT3 phosphorylation.**

A, B    Effect of Gln deprivation on glucose uptake (A) and lactate secretion (B) fluxes in OVCAR3, OVCA420, SKOV3, SKOV3ip cells.

C, D    Influence of AOA, α-ketoglutarate (α-KG) on glucose uptake (C) and lactate secretion (D) fluxes in OVCA cells.

E, F    Effect of glucose deprivation on Gln, glutamate, aspartate and alanine uptake/secretion fluxes in high-invasive (F) and low-invasive (E) OVCA cells.

G       AG490's effect on tyrosine and serine phosphorylation of STAT3 in low-invasive (OVCAR3) and high-invasive (SKOV3) OVCA cells.

H       AG490 shifts low-invasive OVCA cells from glycolysis to OXPHOS, whereas reverse was true for high-invasive cells.

I       AG490 increases oxygen consumption rate (OCR) for OVCAR3 and decreases OCR for SKOV3 cells.

J       AG490 decreases extracellular acidification rate (ECAR) for OVCAR3 and increases ECAR for SKOV3 cells.

K, L    AG490 decreases glucose uptake (K) and lactate secretion rate (L) for OVCAR3 and increases glucose uptake and lactate secretion rate for SKOV3.

Data information: Data in (A–D) are expressed as mean ± SEM, $n \geq 9$. \*\*\*$P < 0.001$. Data in (E, F) are expressed as mean ± SEM, $n = 6$. \*$P < 0.05$, \*\*$P < 0.01$. Data in (H–L) are expressed as mean ± SEM, $n \geq 6$, \*\*\*$P < 0.001$.

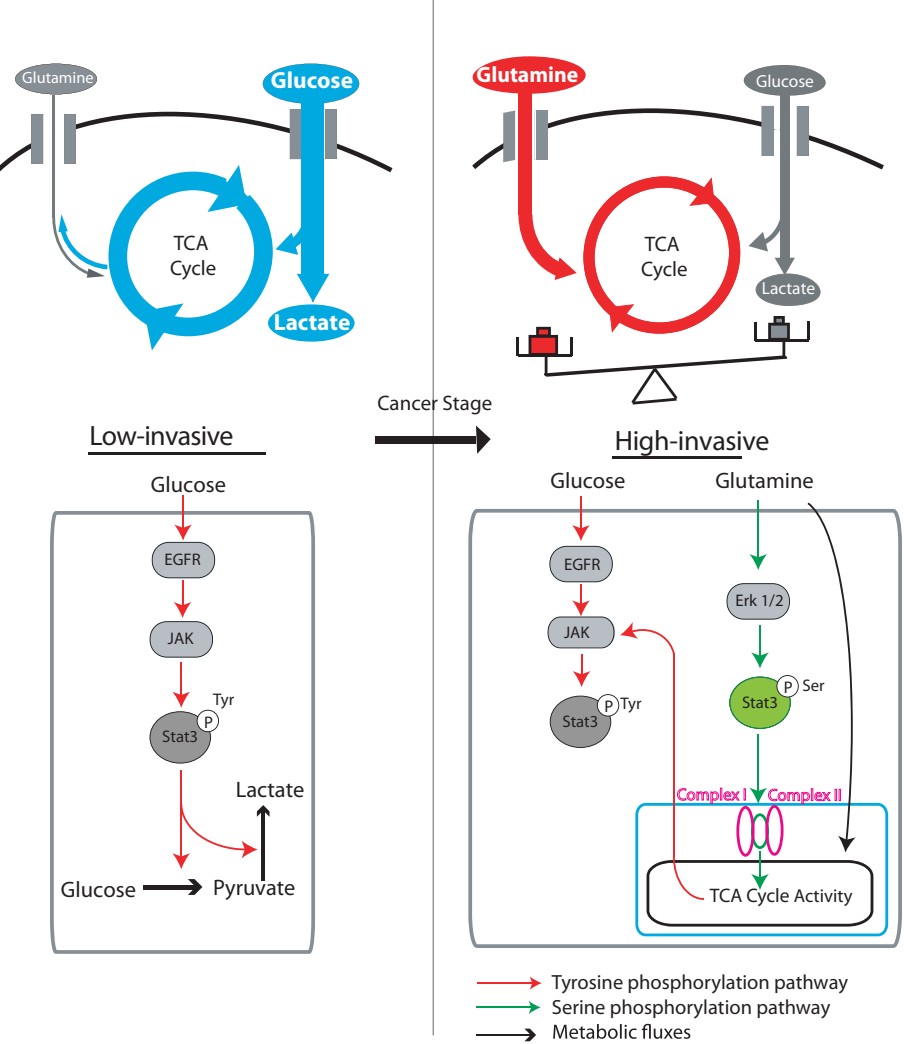

**Figure 8.  Glutamine's entry into tricarboxylic acid (TCA) cycle regulates ovarian cancer (OVCA) invasiveness.**
Schematic showing the shift in nutrient utilization in TCA cycle with increasing degree of invasiveness. Low-invasive OVCA cells are glucose dependent for their TCA cycle pool. With increasing invasiveness in cancer cells, dominant nutrient which feeds the TCA cycle shifts from glucose to Gln. In high-invasive OVCA cells, Gln dominates the TCA cycle. In low-invasive OVCA cells, glucose activates Jak1, which activates STAT3 by tyrosine phosphorylation, thereby regulating glycolysis in cancer cells. In high-invasive OVCA cells, besides glucose's role in activating STAT3 tyrosine phosphorylation, glutamine activates JAK1 through TCA cycle to further activate STAT3 by tyrosine phosphorylation and thus regulate glycolysis. Further, Gln activates Erk1/2, which subsequently activates STAT3 by serine phosphorylation selectively in high-invasive OVCA cells. The serine phosphorylation of STAT3 enhances oxidative phosphorylation in mitochondria by interaction with mitochondrial complexes I and II, thereby increasing TCA cycle activity in high-invasive OVCA cells.

tyrosine phosphorylation of STAT3, whereas Gln deprivation leads to decreased STAT3 serine phosphorylation levels through reduction in Erk 1/2 phosphorylation level, validated by addition of Erk1/2 phosphorylation inhibitor PD98059 (Supplementary Fig S8).

Having established that Gln deprivation reduces STAT3 phosphorylation levels and cellular growth, we found that overexpression of STAT3 can restore reduced proliferation that was observed under Gln deprivation conditions. Based on our results, we postulate that Gln regulates mitochondrial respiration through selective serine phosphorylation of STAT3 in high-invasive OVCA cells. Under Gln deprivation conditions, serine phosphorylation of STAT3 is reduced, which in turn downregulates mitochondrial respiration and increases glycolysis through compensatory pathways. In low-invasive cancer cells, we concluded that respiration is not regulated

through serine phosphorylation, since we did not find expression of serine phosphorylated STAT3 even under complete media conditions. Also in low-invasive OVCA cells, both STAT3 serine and tyrosine phosphorylation pathways are insensitive to Gln deprivation; hence, Gln does not regulate Glc uptake. Glc deprivation decreases tyrosine phosphorylation of STAT3 in OVCA cells, thereby activating the signaling pathways downstream of RTKs, which include Akt phosphorylation and MAPK p38 phosphorylation (Supplementary Fig S6G; Komurov *et al*, 2012). This upregulates the metabolism of other available nutrients, including Gln and serine. For high-invasive OVCA cells, inhibiting Jak with AG490 decreases serine phosphorylation of STAT3 and thus decreases mitochondrial respiration (OCR). Further, compensatory pathways enhance glycolysis by increasing Glc uptake in SKOV3 cells. In low-invasive cells, AG490

marginally increases tyrosine phosphorylation of STAT3. We observe increased respiration and decreased glycolysis in low-invasive cells when STAT3 is inhibited. Future studies are needed to determine the mechanisms by which Gln deprivation negatively regulates STAT3 phosphorylation.

Finally, our work provides insights into the observation that low-invasive OVCA cells are Gln independent. In low-invasive, but not in high-invasive OVCA cells, glucose is converted into glutamine to meet its glutamine requirements. We provide evidence, using galactose and low Glc medium conditions, that low-invasive cells shift their energy metabolism to meet energetic demands. Our work extends this concept by showing that restricting *de novo* Gln synthesis may be effective in targeting low-invasive OVCA cells. A combined approach of targeting high-invasive OVCA cells by blockading Gln entry into TCA cycle pathways, along with targeting low-invasive cancer cells by inhibiting Gln synthesis and STAT3, may provide opportunities for addressing heterogeneity in tumors.

# Materials and Methods

### Cells and reagents

Ovarian cancer cells OVCAR3 and SKOV3 were purchased from ATCC, and OVCAR8 was purchased from NCI on behalf of Rice University. OVCA429, OVCA420 and SKOV3ip cells were obtained from ovarian cell line core at MD Anderson Cancer Center. Cells were grown in RPMI 1640 (10% FBS, pyruvate free, 100 U/ml penicillin and streptomycin). Cells used in these experiments were cultured below 75 passages.

### Cell proliferation

The cells were seeded in 96-well plates overnight, and medium was replaced with specific conditions (regular medium, or medium with Gln deprivation/Glc deprivation, or medium with different drug conditions) for 24, 48 and 72 h. The cell numbers were measured spectrophotometrically by using cell counting kit-8 (*Dojindo*) at 450 nm.

### Clonogenic assay

The cells (300 cells) were seeded in six-well plates overnight, and medium was replaced with specific conditions for 1 week. Colonies were fixed and stained with 0.5% crystal violet and 8% glutaraldehyde for more than 30 min. Wells were washed with water and colonies were counted.

### Matrigel invasion assay

Invasion assays were conducted using BD Matrigel culture inserts. Briefly, 24-well 8.0-μm pore size polyethylene terephthalate membrane inserts (BD Biosciences) were washed twice with RPMI medium and then coated with 20 μl of reduced growth factor Matrigel (1:6 dilution; BD Biosciences) and incubated for 30 min in a 5% CO2 incubator for gel formation. OVCA cells were trypsinized, and 100,000 cells in 200 μl of fresh medium were plated into the upper chamber. Next, 300 μl of medium was added to the lower chamber, and the plate was incubated for 20 h. After incubation, medium in

the lower chamber was aspirated, and invaded cells were treated with 5% glutaraldehyde in PBS for 15 min to fix the cells and then washed in PBS solution three times. Next, 0.5% toluidine blue in 2% sodium carbonate was added to cells for 20 min at room temperature. Subsequently, inserts were washed three times in PBS solution. The noninvaded cells on the inner surface of upper chambers were carefully removed by using a cotton swab. Finally, invaded cells were counted under 20 × magnification for at least three fields per insert.

### Migration wound-healing assay

OVCA cells were seeded in 6-well plates. After confluency, cell monolayers were scratched using a 200-μl pipette tip, washed and cultured in different medium conditions. Images were taken 12/24 h and processed using Image J software (National Institutes of Health).

### Metabolic footprinting

*Glucose consumption assay*
Glucose consumption assay was performed using Wako Glucose kit according to the manufacturer's protocol. Briefly, a 2-μl sample and 250 μl of reconstituted Wako glucose reagent were added to a 96-well assay plate and incubated while shaking at 37°C for 5 min. The change in absorbance, indicating the presence of glucose, was measured at 505 nm by using a spectrophotometer (SpectraMax M5; Molecular Devices).

*Lactate secretion*
Lactate secretion assay was determined using the Trinity Lactate Kit according to manufacturer's protocol. Briefly, lactate reagent was reconstituted with 10 ml of milliohm water and diluted 1:4 in 0.1 M Tris solution (pH 7.0). Media samples were diluted 1:10 in PBS, and lactate reagent was added to the diluted samples in an assay plate. The plate was protected from light and incubated for 1 h before the absorbance was read on a spectrophotometer at 540 nm.

*Amino acid uptake*
Ultra-high-performance liquid chromatography was used to assess amino acid uptake and secretion using Waters Acquity UPLC device. Briefly, media samples were deproteinized, and MassTrak Reagent was added to the samples, along with Borate Buffer/NaOH. Samples were then heated and analyzed using the Waters ACQUITY UPLC system. Eluents were prepared according to Waters' protocol. MassTrak AAA eluent A concentrate was diluted 1:10 in milliQ-water, and MassTrak AAA eluent B was inputted in undiluted form. Flow rate of eluents was 0.4 ml/min, and UV detection was at 260 nm.

### Stable isotope analysis using GC-MS

*Metabolite extraction*
OVCA cells were cultured with a 1:1 mixture of U-$^{13}C_6$ glucose and 1-$^{13}C_6$-labeled glucose, or U-$^{13}C_5$ Gln in 6-well plates for 24 h (DeBerardinis *et al*, 2007; Kornfeld *et al*, 2012). Medium was aspirated, and cells were quenched with 0.4 ml cold methanol, and 0.4 ml ice autoclaved water containing 1 μg norvaline internal standard was

added. Then, cells were collected by scraping with a cell scraper. 1.6 ml of chloroform was added and the cells were vortexed at 4°C for 30 min. Samples were centrifuged at 3000 *g* for 15 min, 4°C.

### Derivatization

Dried samples were dissolved in 50 μl of 2% methoxyamine hydrochloride in pyridine (Pierce) and held at 37°C for 2 h. After the reaction, 45 μl MBTSTFA+1% TBDMCS (Pierce) was added and samples incubated at 55°C for 1 h.

### GC/MS measurements

GC/MS analysis was performed using an Agilent 6890 GC equipped with a 30-m DB-35MS capillary column connected to an Agilent 5975B MS operating under electron impact ionization at 70 eV, and 1 μl of sample was injected in split mode at 270°C, using helium as the carrier gas at a flow rate of 1 ml/min. The GC oven temperature was held at 100°C for 3 min, increased to 250°C at 8°C/min and then to 300°C at 40°C/min and held for 3 min for a total run time of 26 min. Data were acquired in scan mode. MIDs are obtained for each measured metabolite and incorporated with extracellular flux measurements for flux determination. To determine relative metabolite abundances, the integrated signal of all potentially labeled ions for each metabolite fragment was normalized by the signal from norvaline and the per well cell number (obtained by counting surrogate plates). To determine the mass isotopomer distribution (MID) value, each isotopomer's signal is normalized by the sum of isotopomers signals.

## Survival network

Cox proportional hazard regressions were run on the TCGA ovarian data using the 'survival' package in R. Tumor gene expression data and overall survival data from the TCGA ovarian cancer cohort ($n = 360$, 192 events) were used for the survival analysis. The computed hazard ratio was used to color the genes whose products are known to catalyze each reaction step in the glucose and Gln metabolic network.

## Survival analysis

Survival analysis was done on progression-free survival data of OVCA patients obtained from cBioPortal for Cancer Genomics (http://www.cbioportal.org). Gene expression values used were reported from Affymetrix U133A Microarray data in the cBioPortal database. Patient data ($n = 539$) with valid gene expression levels were used to estimate medians and bounds for upper and lower quartiles. Patients were categorized into two groups based on whether the values of gene ratios were above the upper quartile bound and below the lower quartile bound. GLS1 and GLUD1 expression levels were normalized with their respective median values before calculating the (GLS1 + GLUD1)/GLUL ratio. Kaplan–Meier survival graphs were plotted, and log-rank tests were performed using SigmaPlot.

## Ovarian cell line microarray

The KyotoOv38 OVCA cell line gene expression dataset (GSE29175) was downloaded from the Gene Expression Omnibus. Gene expression was normalized relative to the mean expression over all cell line arrays. The relative expression of Gln pathway genes involved in tumor invasion is displayed graphically by heatmap with cell lines ranked by order of invasiveness.

## GLS gene silencing by small interfering RNA

Small interfering RNA (siRNA) targeted to *GLS1* was purchased from Sigma. *In vitro* transient transfection was performed as described previously (Landen *et al*, 2005). Briefly, cells were transfected with *GLS1*-specific or scrambled (control) siRNA using lipofectamine reagent (Invitrogen, Carlsbad, CA). At selected time intervals, cells were harvested to measure mRNA levels of *GLS1* using quantitative reverse transcriptase–PCR (qRT–PCR).

## Quantitative real-time RT–PCR

Total RNA was isolated using Qiagen RNAeasy kit (Qiagen). cDNA was synthesized using the Superscript™ First-strand kit (Invitrogen) as per the manufacturer's instructions. Real-time qRT–PCR was performed to validate the expression of GLS1 in SKOV3ip1 OVCA cells using an 7000 Sequence Detection System (Applied Biosystems, Foster City, CA) with the SYBR GreenER™ qPCR Supermix kit (Invitrogen) as per the manufacturer's instructions. Primers for GLS1 used were:
GLS 1 Forward Primer: GCTGTGCTCCATTGAAGTGAGLS 1 Reverse Primer: GCAAACTGCCCTGAGAAGTC

## *In vivo* models and tissue processing

Female athymic nude mice were purchased from the National Cancer Institute, Frederick Cancer Research and Development Center (Frederick, MD), and maintained according to guidelines set forth by the American Association for Accreditation of Laboratory Animal Care and the US Public Health Service policy on Human Care and Use of Laboratory Animals. All mouse studies were approved and supervised by the MD Anderson Cancer Center Institutional Animal Care and Use Committee. All animals used were between 8 and 12 weeks of age at the time of injection. For cell injection, IGROV1 or SKOV3ip1 cells were trypsinized, washed and suspended in Hanks' balanced salt solution (HBSS; Gibco) before injection. To establish the tumors, $2.0 \times 10^6$ cells 100 μl HBSS cells were injected subcutaneous (s.c.) over the posterior flank. To assess the effect of GLS1 siRNA therapy on tumor growth, treatment with siRNA (150 μg/kg i.p. twice weekly) was initiated 3 days after injection of tumor cells. Mice were randomly divided into two groups ($n = 5$ mice per group) and treated with control and GLS1 siRNA incorporated in neutral nanoliposomes (DOPC; i.p. administration; Landen *et al*, 2005; Stone *et al*, 2012): (i) control siRNA-DOPC and (ii) GLS1 siRNA-DOPC. Treatment was continued for 2 weeks. At the time of necropsy, mouse weight, tumor weight and tumor volume were recorded. Tumor tissue was fixed in formalin for paraffin embedding and H & E staining. Pattern of invasion was assessed. The individuals who performed the necropsies, tumor collections and tissue processing were blinded to the treatment group assignments.

## TMA staining

Tissue microarrays were constructed and immunohistochemical staining performed based on our standard protocols (Rosen *et al*,

2005; Zhai *et al*, 2008; Matsuo *et al*, 2014). After approval by the Institutional Review Board, patient tissue microarray blocks were constructed by taking core samples from morphologically representative areas of paraffin-embedded tissues and assembling them on a recipient paraffin block with a precision instrument (Beecher Instruments) as described previously (Merritt *et al*, 2008). GLS1 expression was determined by semi-quantitatively assessing the percentage of stained tumor cells using the protocol described previously (Merritt *et al*, 2008). Briefly, the stained slides were scored on the basis of their immunohistochemical score (high expression, >100; low expression, ≤100; McCarty *et al*, 1985; Singh *et al*, 2007).

### XF bioenergetics assay

Mitochondrial OCR and ECAR were conducted using a Seahorse XF24 Analyzer. The cells were seeded in Seahorse 24-well microplates and cultured at 37°C with 5% CO2. The following day, the media was replaced with 700 µl of assay medium composed of RPMI without FBS and sodium bicarbonate and incubated at 37°C without $CO_2$ for 1 h. The basal OCR and ECAR were measured. To measure cells' mitochondrial capacity, drugs were injected to the final concentration as 2 µg/ml of oligomycin, 2.5 µM of carbonylcyanide-ptrifluoromethoxyphenylhydrazone (FCCP) and 2 µM of antimycin A. To measure cells' glycolytic capacity, drug was injected to the final concentration as 11 mM glucose, 20 mM 2-DG.

### ATP measurements

The intracellular ATP content was measured for the different cell types using the Cell Titer-Glo Luminescent cell viability assay (Promega). The cells were seeded in white 96-well plates at 10 K cells per well. The following day, the media were replaced with specific medium (complete RPMI, RPMI without Gln/glucose) and incubated for 24 h. For oligomycin's effect on intracellular ATP, the cells were then incubated for 7 h by using RPMI with different concentration of oligomycin. The ATP content was thereby measured according to the manufacturer's instructions, and ATP content was normalized with cell number.

### Mitochondrial membrane potential assay

Cells were seeded in 96-well plates to 90% confluency. Medium was replaced with medium with 0.03% $H_2O_2$ (with Gln/without Gln, with glucose/without glucose) for 1 h. Then, wells were loaded with tetramethylrhodamine methyl ester (TMRM) in 37°C for 15 min and washed with PBS twice to remove residue TMRM. Values were read at 548/574.

### Reactive oxygen species assay

Cells were seeded in 96-well plates to 90% confluency. Fresh medium with $H_2$DCFDA was loaded in 37°C for 30 min. Remove cell-permeant 2′,7′-dichlorodihydrofluorescein diacetate ($H_2$DCFDA) medium and wash twice with PBS. PBS with 0.03% $H_2O_2$ (with Gln/without Gln, with glucose/without glucose) was loaded and the fluorescence values at 485/538 after 1 and 2 h were read.

### NADPH measurements

66.6 µl 1-methoxy N-methyl dibenzopyrazine methyl sulfate (PMS: 0.5 mM in DMSO) and 3.2 ml 2,3-bis-(2-methoxy-4-nitro-5-sulfophenyl)-2*H*-tetrazolium-5-carboxanilide (XTT, 251 mM in DMSO) was mixed to prepare XTT/1-methoxy PMS solution. Cells were seeded in 96-well plates. Different fresh medium (with Gln/without Gln, with glucose/without glucose) was loaded for 2 h and replaced with corresponding medium with 0.2% XTT/1-methoxy PMS solution for another 1 h, and absorbance value at 450 nm with 650 nm was measured as the reference.

### NADPH/total NADP measurements

Cells were seeded in six-well plates overnight and replaced with specific medium (RPMI, RPMI without Gln or glucose) for 24 h. The cells were lysed. The lysed samples and cycling buffer (100 mM Tris–HCl, 0.5 mM MTT, 2 mM PES, 5 mM EDTA) were mixed and added to clear 96-well plates. For NADPH extraction, we heated samples for 30 min at 60°C. 1.6 U of glucose-6-phosphate dehydrogenase was added to each well, and the plates were incubated in the dark for 30 min. Afterward, 10 mM glucose-6-phosphate was added to each well, and absorbance was read at 570 nm each minute for 5 min.

### GSH/GSSG measurements

The GSH/GSSG assays were conducted according to the manufacture's protocol of Promega GSH/GSSG Glo™ assay kit. Briefly, cells were seeded in white 96-well plates overnight, and medium was replaced with different fresh medium (with Gln/without Gln, with glucose/without glucose) for 3 h. Medium was removed and cells were lysed with either total or oxidized glutathione reagent (50 µl). Then plates were shaked for 5 min and added with 50 µl luciferin generation reagent to incubate 30 min at room temperature. 100 µl of luciferin detection reagent was added into well, and plates were shaked briefly, equilibrated for 15 min and read luminescence.

### Transfection

Prior to transfection, $2 \times 10^5$ SKOV3 cells were plated in 6-well plate in 1 ml of RPMI + 10% FBS without antibiotics. The cells were transfected with 2.5 µg DNA and 5 µg lipofectamine 2000 for 16 h. The transfected cells were seeded in to 96-well plates and treated with experimental media conditions. Cellular proliferation was measured at 24-h timepoints using the Cell Counting Kit-8 reagent (Enzo life sciences).

### Western blot

Following treatment with drugs and different media conditions, cells were washed and then lysed for 20 min on ice with lysis buffer [50 mM HEPES pH 7.4, 150 mM NaCl, 1 mM ethylene glycol tetraacetic acid (EGTA), 10 mM sodium pyrophosphate, 100 mM NaF, 1.5 mM MgCl2, 10% (v/v) glycerol, 1% (w/v) Triton X-100, protease inhibitor cocktail (Sigma) and phosphatase inhibitor (Sigma)]. Cell debris was removed by centrifugation, and the total protein content

of the cell lysate was determined by Bradford assay. Five to 50 μg of total protein was separated by SDS–PAGE and subsequently blotted to nitrocellulose membranes. Protein and phospho-protein levels were probed with diluted primary antibodies (Cell Signaling) and HRP-conjugated secondary antibodies (Thermo Scientific) and detected by chemiluminescence and exposure to autoradiography film.

### TCGA data

The microarray gene expression data from patient tumor samples made publically available by TCGA were downloaded and analyzed for the relative expression of metabolic genes. Patient tumor samples were annotated based on the clinical data made available and were grouped based on whether the patients presented with venous or lymphatic tumor invasions. The relative levels of gene expression were analyzed using a two-tailed Student's *t*-test.

### Ovarian cell line microarray

The KyotoOv38 OVCA cell line gene expression dataset (GSE29175) was downloaded from the Gene Expression Omnibus. Gene expression was normalized relative to the mean expression over all cell line arrays. The relative expression of glutamine pathway genes and Stat3 target genes involved in tumor invasion is displayed graphically by heatmap with cell lines ranked by ordered of invasiveness.

### Statistical analysis

Some statistical analyses were performed using Student's *t*-test, and these data were reported in bar graphs as means ± SEM. For multiple comparisons, one-way ANOVA with Tukey's tests was used for statistical analysis.

**Supplementary information** for this article is available online: http://msb.embopress.org

### Acknowledgements

This work made possible in part through support from the Ken Kennedy institute for Information technology at Rice University to D.N. under the Collaborative Advances in Biomedical Computing 2011 seed funding program supported by the John and Ann Doerr Fund for the Computational Biomedicine. Funding as an Odyssey Fellow to T.J.M. was supported by the Odyssey Program and Estate of C. G. Johnson, Jr. at The University of Texas MD Anderson Cancer Center. Portions of this work were supported by the NIH (CA016672, P50 CA083639, P50 CA098258, U54 CA151668, UH2 TR000943-01), CPRIT (RP110595), the Ovarian Cancer Research Fund, Inc. (Program Project Development Grant), the Blanton-Davis Ovarian Cancer Research Program, the Gilder Foundation, the Betty Anne Asche Murray Distinguished Professorship to A.K.S. We acknowledge support from NIH R21NS074151 (NINDS) to T.T.

### Author contributions

Study concept and design: DN and LY; Acquisition of data: LY, TM, LSM, JM, HZ, SW, GA, DJ, AA, JW, RR, CR, IM, GL, JL, TT; Analysis and interpretation of data: LY, TM, LSM, AKS, PTR, DN; Drafting of the manuscript: LY and DN; critical revision of the manuscript for intellectual content: LY, AKS, LSM, PTR, DN; obtained funding: AKS, PTR, DN; administrative, technical or material support: AKS, PTR, DN; all authors contributed to the preparation of the manuscript.

### Conflict of interest

The authors declare that they have no conflict of interest.

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
