## [Review Process File · Molecular Systems Biology]

Metabolic shifts toward glutamine regulate tumor growth, invasion and bioenergetics in ovarian cancer

Lifeng Yang, Tyler Moss, Lingegowda S. Mangala, Juan Marini, Hongyun Zhao, Stephen Wahlig, Guillermo Armaiz-Pena, Dahai Jiang, Abhinav Achreja, Julia Win, Rajesha Roopaimoole, Cristian Rodriguez-Aguayo, Imelda Mercado-Uribe, Gabriel Lopez-Berestein, Jinsong Liu, Takashi Tsukamoto, Anil K. Sood, Prahlad T. Ram, and Deepak Nagrath

Corresponding author: Deepak Nagrath, Rice University

Review timeline:

Submission date:	09 October 2013
Editorial Decision:	06 November 2013
Revision received:	07 February 2014
Editorial Decision:	12 March 2014
Revision received:	18 March 2014
Accepted:	25 March 2014

Editor: Maria Polychronidou

Transaction Report:

1st Editorial Decision

06 November 2013

Thank you again for submitting your work to Molecular Systems Biology. We have now heard back from the two referees who agreed to evaluate your manuscript. As you will see from the reports below, while the reviewers acknowledge that the presented findings are potentially interesting, they raise a series of concerns, which should be convincingly addressed in a revision of the manuscript.

Without repeating all the points listed below, among the more fundamental issues raised by the referees are the following:

- As pointed out by reviewer #2, inclusion of in vivo experiments would significantly strengthen the main conclusion of the study, namely the link between metabolism and tumor aggressiveness.
- The correlation between proliferation rate and invasiveness should be examined.

If you feel you can satisfactorily deal with these points and those listed by the referees, you may wish to submit a revised version of your manuscript. Please attach a covering letter giving details of the way in which you have handled each of the points raised by the referees. A revised manuscript will be once again subject to review and you probably understand that we can give you no guarantee at this stage that the eventual outcome will be favorable.

Reviewer #1

In this work, the authors examined a panel of ovarian cancer cell lines, and through experiments

examining their clonogenic and invasive potential as well as their metabolic behavior, they identified a strong correlation between aggressiveness and glutamine dependence. Furthermore, through statistical analysis of publicly available microarray data from ovarian cancer patients, they identified a correlation between poor survival and the transcription of genes pertaining to glutamine catabolism (glutaminase and glutamate dehydrogenase) relative to those pertaining to glutamine synthesis (glutamine synthetase). Additional experiments revealed a selective reliance of the aggressive (glutamine-dependent) cell lines on glutamine for maintenance of mitochondrial potential, generation of reducing equivalents, and production of ATP. Western blot analyses demonstrated activation of a number of oncogenic signaling proteins, most dramatically STAT3, are concurrently activated with glutamine addition in aggressive ovarian cancer cell lines.

In the eyes of the reviewer, the findings of this paper are of great interest. Clonogenic capability and invasiveness are fundamental components of malignancy, and the connection that the authors have displayed between these properties and reliance on glutamine (especially in contrast to the role of glucose in the examined cell lines) in the context of ovarian cancer provides valuable insight into the role of metabolism in oncogenic transformation. Further, the connection to patient survival and glutamine catabolism/synthesis gene expression identified may provide a novel biomarker for aggressive ovarian cancer, and the experiments showing STAT3 activation gives some understanding of the underlying molecular mechanism.

However, before the article is approved for publication, the reviewer believes that a number of issues and points that currently limit its clarity. First, there are a large number of grammatical and usage errors throughout the entire manuscript; normally, this would not be a major issue, and the reviewer does not believe that this detracts from the merits of the authors' conclusions, but the frequency of these mistakes presently restricts its comprehensibility and distracts from its impact. An extensive review and correction of these errors should be performed before the article is published. More critically, the manuscript appeared often to alternate between different citation formats, which made validation of some referenced claims impossible at times; it goes without saying that correcting these inconsistencies is essential before publication.

Regarding technical content, a number of concerns that the reviewer feels should be addressed before publishing exist; these include the following:

- At least some of the enzymes for which the authors have evaluated transcriptomic data possess multiple isoforms (e.g. kidney- and liver-type glutaminases [GLS1 and GLS2]), but the reviewer was not able to find information on which forms were used in the authors' analyses in the manuscript. The authors should explicitly mention these if they have not done so, and if they have, they should more clearly state this information (e.g. within the relevant section of the manuscript's main body, figure legend, or methods section).
- The authors' data demonstrate a positive correlation between expression of genes involved in glutamine catabolism and poor survival result, but a negative correlation between expression of genes involved in glucose transport and glycolysis and poor survival outcome. Though they convincingly show that glutamine is required for proliferation and metastatic potential in aggressive ovarian cancer cells in a manner that is distinct from glucose, the notion of glucose transporter and glycolytic enzyme expression being correlated negatively with poor survival appears to contradict a number of prior studies which appear to have concluded the opposite in other cancers [e.g. Kunkel et al. (2003). *Cancer*, 97(4), 1015-24; Haber et al., (1998). *Cancer*, 83(1), 34-40]. The reviewer would appreciate if the authors acknowledge these results and perhaps discuss potential explanations for this discrepancy (e.g. May this phenomenon be specific for ovarian cancer? Are these aggressive ovarian cancer cells simply a subset that has specifically developed mechanisms for glutamine to compensate some of glucose's traditionally-recognized goals?).
- The authors should more clearly justify why they used a 1:1 combination of [U-13C6]glucose and [1-13C]glucose rather than simply 100% [U-13C6]glucose to determine glucose contribution to specific metabolites. (Presumably, this is to allow for simultaneous estimation of relative oxidative PPP activity by 1-13C loss, but again, this should be clearly stated.) As well, somewhere in the manuscript (e.g. methods, figure captions, or supplemental information), the authors should clearly

state how they normalized their metabolite labeling results to properly assess glucose contribution with this tracer distribution (the authors appear to have done this, and if they haven't then they should have when appropriate). This includes any figures or values resulting from this data, such as Figs. 3H-I.

- On p. 12, in reference to Sup. Fig. 3A, the authors mentioned that in aggressive SKOV3 cells, other amino acids such as asparagine, serine, and glycine were unable to rescue the reduction in proliferation induced by glutamine depletion, but according to the methods section, these cells were all grown in RPMI 1640, which already contain each of these amino acids. Were these amino acids normally not in the media here? In the proliferation experiments, were the authors using media lacking nonessential amino acids unless otherwise state? Again, the authors should provide a clarification.

- Fig. 3I suggests that PPP flux is elevated in the non-aggressive cells vs. aggressive cells, but the authors were not specific about how "PPP Pathway" activity was quantified. On p. 36, it is suggested that the M1 and M0 isotopomers of lactate were used to estimate this, but the authors were not specific (e.g. using equations or formulas) how they did this. Somewhere in the manuscript (e.g. methods), the authors should clearly define this. In addition, though the authors simply use the term "PPP flux," they appear to be exclusively referring the oxidative PPP. If this is the case, they should specify that it is, and if it isn't, they should explicit in how they are defining the term.

- To explain their experiments investigating signaling pathways, the authors have proposed a model in which, in aggressive cells, glutamine activates STAT3 by both serine and tyrosine phosphorylation downstream of its presence (see Fig. 7). However, many of the connections that they try to make to explain this process appear to be potentially incompletely substantiated. For instance, while both serine phosphorylation of STAT3 and ERK1/2 phosphorylation appeared to occur concurrently following glutamine addition, the reviewer finds it difficult to conclude that glutamine mediates phosphorylation of STAT3 S727 via ERK1/2 without an experiment showing a reduction in this activity using some means to validate causality, such as showing a reduction in STAT3 S727 phosphorylation in response to an ERK inhibitor or a reduction in functional ERK levels. Unless such evidence to distinguish causation is available, it may be preferable to weaken claims such as the interactions shown between signaling proteins on the Fig. 7 schematic. (Again, it is possible that some of the references made when discussing this portion demonstrate such causal relations, but the citations in this section were linked to a numerical format that was incompatible with the subsequent bibliography. Even if this is the case, a more thorough discussion of how this causality was established should be given.)

- In Fig. 5C, a protein called Stat3c is expressed, and in Fig. 5I, a compound called "Sttatic" is used; these appear to respectively refer to a constitutively active form of STAT3 and an inhibitor of STAT3, as judging by the figure captions, but nowhere is this clearly stated. These conditions should be clearly defined in the relevant sections in the main text when they are discussed, as should all abbreviations or terms that are not widely known.

- This note may be minor, but on p. 23, the authors mention "In poorly aggressive OVCA cells, glutamine contributed more than 50% of pyruvate... [o]n the other hand, our results demonstrate that in aggressive OVCA cells, glutamine contributed to more than 50% of glutamate..." While it is clear that the authors intended the comparison in TCA cycle metabolite enrichment (excluded here) to show the differential contribution of glucose and glutamine to between each class of cell lines, it is not clear why they compared the glucose-to-pyruvate and glutamine-to-glutamate contributions to one another between different lines. This should be further justified; otherwise, such a comparison should not be made without including additional information (e.g. aggressive cell glucose-to-pyruvate contribution and non-aggressive cell glutamine-to-glutamate contribution).

- For any conclusions drawn regarding relative metabolic flux activity from labeling data, the observed system must be at isotopic steady state (i.e. labeling shouldn't change with time), but this did not appear to be discussed in the methods section. The authors should show that this does indeed hold for their system when they have drawn such conclusions (or at least cite references that

establish or suggest that their conditions should be sufficient).

- It is unclear to the reviewer what the authors mean in the figures when the y-axis units on growth curves are given as "Relative Proliferation Normalized with RPMI." How is this term defined? Is the cell count on each given day divided by the corresponding cell count on the same day for cells grown in complete media? Terms such as this (including the normalized glucose uptake and lactate secretion rates in Fig. 6) need to be clearly stated.

- As well, in the Fig. 3 caption, the sentence "OCR was normalized with value before injections" appears; again, it was unclear what this meant, and this normalization condition should be clearly defined.

- In Fig. 4J, the units on the y-axes are given as AU, which is acceptable, but what these arbitrary units are representing or are proportional to needs to be clearly stated. (Presumably, judging by the caption, this is cell number, but this should be shown explicitly somewhere, ideally on the figure.)

- Loading control proteins should be provided on Figs. 5C-E, as they are for other Western Blots.

- In estimating glycolytic/ETC pathway contribution to ATP synthesis using oligomycin, as was done for Fig. 2K, even if total ATP levels stay the same following oligomycin-mediated inhibition of ATP synthetase (as was shown), the notion that glycolysis will completely compensate for the disruption in respiration should only hold true if the total rate of ATP consumption remains unchanged, which seems difficult to validate. In using these measures to infer percentage contribution to ATP content, the authors should cite a source indicating that it is a valid measure as such; otherwise, they should label their data to more accurately reflect what was directly measured.

- In Fig. 3H, it is unclear why glucose-labeled and glutamine-labeled pyruvate and lactate are being compared to one another; these values measure completely different things (and would normally be expected to be very different), and indeed the differences between each condition appear to be very similar (with the minimum p-value for each). The purpose behind this figure and such a comparison should be better explained, and perhaps the statistical significance/p-value icons should be removed unless their purpose is otherwise necessary.

- At some point in the manuscript (say either the main body or the supplemental figure captions), the authors should explain the significance of each of the protein levels and phosphorylation statuses examined in Fig. S5F is relevant. (E.g. The manuscript mentioned that some are indicating autophagy, but did not say which one(s).)

- Although they don't need to include the descriptions in the methods and materials section, they authors should at least mention that the Seahorse, ATP measurements, ROS measurement assays, and all other protocols are given in the supplementary methods section (e.g. just one sentence in the methods and materials stating the additional protocols given in the supplemental section).

- In the Supplementary Information section, the authors used a lot of unfamiliar abbreviations (e.g. TMRM, XTT) that were never clearly defined. The authors should define each abbreviation first before using it.

As indicated, the reviewer believes this manuscript should be published, but not until each of the previously-mentioned points is addressed.

Reviewer #2

In this manuscript Yang et al showed that ovarian cancer cells have distinctive glutamine metabolic traits which correlate with tumor's invasive capacity. The study is original, and the study of

glutamine metabolism is comprehensive. However, the relation between metabolism and aggressiveness of the tumor remains speculative and should not be emphasized in the title and in the text.

In particular, the assumption that the in-vitro invasive capacity of the cells, assessed by a transwell migration assay, is readout for aggressiveness of the tumor is not supported by experimental evidences. An in-vivo approach would help clarifying this basic point and strengthen the study. To support the results obtained with the cell lines, the authors analyzed transcriptional profiles of ovarian tumor samples. They concluded that the ratio between *GLUD1* and *GLUL* can predict patients' prognosis. However, it is known that the expression of glutamine metabolic genes can be regulated post-translationally, in particular, Glutamine Synthetase protein abundance is controlled by protein degradation in many tissues. Thus the potential of these biomarkers for aggressiveness should be validated at the protein levels in the cell lines, and more importantly, in samples from ovarian tumor tissues. The authors should also comment on the opposite 'hazard ratios' found for genes of the glutaminolytic pathway (*GLS* and *GLUD*) and the glutamine transporters (*SLC1A5*, *SLC38A1*). This is in particularly unclear as the glutamine transporters feed the reactions catalyzed by the glutaminolytic enzymes (Fig 1G).

One important factor which contributes to tumor aggressiveness is the cell proliferative potential. The results presented in this study do not actually compare the proliferation rates of the cell lines, only the decrease in proliferation under starved conditions. The possible correlation between the basal proliferation rate of the cells and the invasive ability should be assessed. It is possible that higher proliferation rate leads to both, higher dependency on glutamine and higher invasive potential; hence, the correlation between glutamine dependency and invasiveness is secondary to the proliferation rate of the cells.

The study characterized the metabolic fate of glutamine and glucose carbons in representative cell lines (Fig 2 and 3). In this context, the interpretative advantage of using a mixture of U-13C6 and 1-13C glucose is questionable and more likely to generate confusion due to isotomer mixing from different sources. Moreover the activity of malic enzyme (as shown in Fig 3) which produces glutamine-derived 13C3 pyruvate is not taken into account in Figure 2. This affect the potential isotopomer distribution represented in scheme 2E and the conclusions thereafter.

The authors refer to the exchange of metabolites between cells and the medium (Fig 2 and 6) as uptake/secretion rate. However, the time unit for this rate is not shown. Furthermore, the normalization of the exchange rates suffers a more fundamental problem: the consumption/secretion rates should take into account the cell growth during the whole experimental period assessed and not just the number of cells at the beginning or end of the study. It is intuitive that faster growing cells consume more nutrients per unit of time than slow ones, independently of their metabolic traits. For this reason, a normalization that takes into account cell growth is necessary to evaluate potential differences in the metabolic phenotype. Without this, the conclusions drawn from such a study are invalid.

In figure 4H, the authors measured mitochondrial respiration in the "glutamine independent" cells in the absence of glutamine and presence of MSO, an inhibitor of Glutamine Synthetase. The results of this experiment are misinterpreted: the decrease in OCR observed upon MSO addition cannot be justified by a supposed decrease in the flux of glutamine into the TCA cycle. In fact, by inhibiting the conversion of glutamate to glutamine, MSO is expected to increase the availability of glutamate for the TCA cycle. The same principle applies for the effect of MSO on glutamine-fed cells, where MSO stimulated cell invasiveness, potentially by favoring glutamate flux into the TCA cycle hence, de-facto increasing glutaminolysis. This however, needs to be assessed by direct tracing of glutamine under MSO-treated conditions.

It would be of interest to assess whether STAT3 phosphorylation is specifically regulated by glutamine or whether it can be reproduced by other non-essential amino acids.

The authors hypothesized that the increase in serine uptake observed during glucose deprivation, could produce glycolytic intermediates. It is unclear (and unexplained) how serine may be utilized by the cells under glucose-starved conditions and the references used by the authors do not provide such explanation.

Response to Editor's comments

Thank you again for submitting your work to Molecular Systems Biology. We have now heard back from the two referees who agreed to evaluate your manuscript. As you will see from the reports below, while the reviewers acknowledge that the presented findings are potentially interesting, they raise a series of concerns, which should be convincingly addressed in a revision of the manuscript.

We thank the reviewer's and the editors for finding our manuscript interesting. We have highlighted in blue all the changes in the manuscript.

Without repeating all the points listed below, among the more fundamental issues raised by the referees are the following:

- As pointed out by reviewer #2, inclusion of in vivo experiments would significantly strengthen the main conclusion of the study, namely the link between metabolism and tumor aggressiveness.
- The correlation between proliferation rate and invasiveness should be examined.

If you feel you can satisfactorily deal with these points and those listed by the referees, you may wish to submit a revised version of your manuscript. Please attach a covering letter giving details of the way in which you have handled each of the points raised by the referees. A revised manuscript will be once again subject to review and you probably understand that we can give you no guarantee at this stage that the eventual outcome will be favorable.

We thank the editors for encouraging us to revise the manuscript. In our revised submission, we have focused our efforts to perform all the experiments necessary to respond satisfactorily to the editor's and reviewer's comments in the shortest time possible.

Legend: R: reviewer's comment, A: author's response, C: Corrections

Response to Reviewer's comments

Reviewer #1

R1: In this work, the authors examined a panel of ovarian cancer cell lines, and through experiments examining their clonogenic and invasive potential as well as their metabolic behavior, they identified a strong correlation between aggressiveness and glutamine dependence. Furthermore, through statistical analysis of publicly available microarray data from ovarian cancer patients, they identified a correlation between poor survival and the transcription of genes pertaining to glutamine catabolism (glutaminase and glutamate dehydrogenase) relative to those pertaining to glutamine synthesis (glutamine synthetase). Additional experiments revealed a selective reliance of the aggressive (glutamine-dependent) cell lines on glutamine for maintenance of mitochondrial potential, generation of reducing equivalents, and production of ATP. Western blot analyses demonstrated activation of a number of oncogenic

signaling proteins, most dramatically STAT3, are concurrently activated with glutamine addition in aggressive ovarian cancer cell lines. In the eyes of the reviewer, the findings of this paper are of great interest. Clonogenic capability and invasiveness are fundamental components of malignancy, and the connection that the authors have displayed between these properties and reliance on glutamine (especially in contrast to the role of glucose in the examined cell lines) in the context of ovarian cancer provides valuable insight into the role of metabolism in oncogenic transformation. Further, the connection to patient survival and glutamine catabolism/synthesis gene expression identified may provide a novel biomarker for aggressive ovarian cancer, and the experiments showing STAT3 activation gives some understanding of the underlying molecular mechanism.

However, before the article is approved for publication, the reviewer believes that a number of issues and points that currently limit its clarity. First, there are a large number of grammatical and usage errors throughout the entire manuscript; normally, this would not be a major issue, and the reviewer does not believe that this detracts from the merits of the authors' conclusions, but the frequency of these mistakes presently restricts its comprehensibility and distracts from its impact. An extensive review and correction of these errors should be performed before the article is published. More critically, the manuscript appeared often to alternate between different citation formats, which made validation of some referenced claims impossible at times; it goes without saying that correcting these inconsistencies is essential before publication.

A1: We thank the reviewer for finding our results to be of significance and helping us address issues that limits the manuscript's impact. Based on the reviewer's suggestion, we have scrutinized the manuscript for all possible grammar and usage errors. We have also rectified the citations to comply with the recommended MSB format. We feel that manuscript has substantially improved after these suggested changes.

R2: At least some of the enzymes for which the authors have evaluated transcriptomic data possess multiple isoforms (e.g. kidney- and liver-type glutaminases [GLS1 and GLS2]), but the reviewer was not able to find information on which forms were used in the authors' analyses in the manuscript. The authors should explicitly mention these if they have not done so, and if they have, they should more clearly state this information (e.g. within the relevant section of the manuscript's main body, figure legend, or methods section).

A2: We thank the reviewer for pointing this out. In the previous version, we had represented GLS1 as GLS (kidney-type glutaminases). To clarify the GLS isoforms, we have changed GLS to GLS1 in the main body, figure legends and methods sections. We did not identify problems with other enzymes involved in the transcriptomics data.

R3: The authors' data demonstrate a positive correlation between expression of genes involved in glutamine catabolism and poor survival result, but a negative correlation between expression of genes involved in glucose transport and glycolysis and poor survival outcome. Though they convincingly show that glutamine is required for proliferation and metastatic potential in aggressive ovarian cancer cells in a manner that is distinct from glucose, the notion of glucose transporter and glycolytic enzyme expression being correlated negatively with poor survival appears to contradict a number of prior studies which appear to have concluded the opposite in other cancers [e.g. Kunkel et al. (2003). *Cancer*, 97(4), 1015-24; Haber et al., (1998). *Cancer*, 83(1), 34-40]. The reviewer would appreciate if the authors acknowledge these results and

perhaps discuss potential explanations for this discrepancy (e.g. May this phenomenon be specific for ovarian cancer? Are these aggressive ovarian cancer cells simply a subset that has specifically developed mechanisms for glutamine to compensate some of glucose's traditionally-recognized goals?).

A3: We believe that the reviewer's observation is correct in this regard. Our findings that glycolytic genes are associated with better patient survival, is in contrast with currently understood mechanisms. In the references that the reviewer mentioned, GLUT1 expression level and glucose metabolism positively correlates with patient mortality in colorectal carcinoma and oral squamous cell carcinoma. Further, there is an increased GLUT1 expression in gastrointestinal carcinoma, breast carcinoma, squamous cell carcinoma of the head and neck, renal cell carcinoma and hepatoma. Hence, we analyzed glutaminolysis/glycolytic metabolic networks' gene expression and survival data from kidney and breast cancer patients from TCGA. In kidney cancer patients we found that only high expression of genes related to glucose metabolism correlate with poor survival. Whereas, in breast cancer patients, genes related to both glucose and glutamine metabolism correlate with poor patient prognosis. This suggests that our observations are specific to ovarian cancer. Since the TCGA database contains information for a large number of patients, which consists of different types of ovarian cancer, we do not believe that the aggressive ovarian cancer cells used in our paper only reflect the phenomena for a subset of ovarian cancer. Based on reviewer's suggestion, we have acknowledged the mentioned references and included the network correlation data obtained from kidney and breast in the supplementary section (correlation between metabolic gene expression and patient outcome for breast cancer patients using TCGA data, Supplementary Figure 2A; and kidney cancer patients using TCGA data, Supplementary Figure 2B). Please see the corrections (C3) below, which were added in the revised manuscript.

C3:

We have added the following correction to illustrate that patient outcome correlation with metabolic genes varies significantly depending on origin of cancer.

Page 7, paragraph 1: Our findings are in contrast with previous studies that show correlation of poor outcome with high expression of glycolysis genes in some cancers (Haber et al, 1998; Kunkel et al, 2003). To confirm if our findings are specific to OVCA, we analyzed the TCGA data from breast and kidney cancer patients. The gene expression network in kidney cancer patients shows that high expression of Glc metabolism genes correlates with poor survival, while in breast cancer patients TCGA data shows that both Glc and Gln metabolism correlate with poor survival (Sup. Figs. 2A-2B).

A

B

R4: The authors should more clearly justify why they used a 1:1 combination of [U-13C6] glucose and [1-13C]glucose rather than simply 100% [U-13C6]glucose to determine glucose

contribution to specific metabolites. (Presumably, this is to allow for simultaneous estimation of relative oxidative PPP activity by 1-13C loss, but again, this should be clearly stated.) As well, somewhere in the manuscript (e.g. methods, figure captions, or supplemental information), the authors should clearly state how they normalized their metabolite labeling results to properly assess glucose contribution with this tracer distribution (the authors appear to have done this, and if they haven't then they should have when appropriate). This includes any figures or values resulting from this data, such as Figs. 3H-I.

A4: We thank the author for bringing this to our attention. Indeed, we used 1:1 combination of [U-13C6] glucose and [1-13C] glucose for simultaneous estimation of relative oxidative PPP activity and glucose metabolism. The oxidative pentose phosphate pathway was estimated using M0 and M1 labeling of lactate in cells cultured with 1:1 mixture of U-13C6 glucose and 1-13C labeled glucose. We have now included the normalization method for metabolites in the methods section. Following are the respective corrections:

C4:

Page 13, paragraph 2: In contrast, poorly aggressive cancer cells had increased oxidative pentose phosphate pathway flux (PPP) compared to aggressive cancer cells (Fig. 3J), which was estimated using M0 and M1 labeling of lactate in cells cultured with 1:1 mixture of U-13C6 Glc and 1-13C labeled Glc. 1-13C labeled Glc loses the labeled carbon when it is routed through oxidative PPP, and then is converted into M0 pyruvate (Gaglio et al, 2011).

Page 30, paragraph 3: To determine the mass isotopomer distribution (MID) value, each isotopomer's signal is normalized by the sum of isotopomers signals.

Page 41, Figure 3H caption: Comparison of M3 pyruvate and M3 lactate derived from either glucose (1:1 mixture of U-13C6 glucose and 1-13C1 glucose) or Gln (U-13C5 Gln) in aggressive and poorly aggressive OVCA cells. Since all conditions have complete media, their total pyruvate and lactate content should be the same.

R5: On p. 12, in reference to Sup. Fig. 3A, the authors mentioned that in aggressive SKOV3 cells, other amino acids such as asparagine, serine, and glycine were unable to rescue the reduction in proliferation induced by glutamine depletion, but according to the methods section, these cells were all grown in RPMI 1640, which already contain each of these amino acids. Were these amino acids normally not in the media here? In the proliferation experiments, were the authors using media lacking nonessential amino acids unless otherwise state? Again, the authors should provide a clarification.

A5: We thank the reviewer for bring this to our attention. We did use RPMI 1640, which contains these nonessential amino acids, but the concentration of these amino acids is relatively low in the media, therefore, we increased the concentration to that indicated in the Sup. Fig 3A. In proliferation experiments, we used the RPMI without glutamine/glucose and did not modify other amino acids. We have clarified now in the revised text.

C5:

Page 12, paragraph 2: The addition of ammonia and increasing concentration of other nonessential amino acids, including aspartate, asparagine, serine, glycine, and alanine failed to rescue the aggressive OVCA cells from Gln depletion (Sup. Fig. 4A).

R6: Fig. 3I suggests that PPP flux is elevated in the non-aggressive cells vs. aggressive cells, but the authors were not specific about how "PPP Pathway" activity was quantified. On p. 36, it is suggested that the M1 and M0 isotopomers of lactate were used to estimate this, but the authors were not specific (e.g. using equations or formulas) how they did this. Somewhere in the manuscript (e.g. methods), the authors should clearly define this. In addition, though the authors simply use the term "PPP flux," they appear to be exclusively referring the oxidative PPP. If this is the case, they should specify that it is, and if it isn't, they should explicit in how they are defining the term.

A6: We appreciate the reviewer's suggestion. We have now included the suggested changes.

C6:

Page 41, Figure 3E caption: Lactate and citrate generation from glucose through direct glycolysis or through both oxidative pentose phosphate pathway (PPP) and glycolysis. Pink circles represent unlabeled carbons from oxidative PPP.

Page 42, Figure 3J caption: Oxidative pentose phosphate pathway fluxes estimated using the mathematical relation (percentage of unlabeled M0 lactate - percentage of labeled M1 lactate) to represent relative oxidative PPP fluxes in OVCAR3 and SKOV3 cells cultured with 1:1 mixture of $^{13}\text{C}_6$ glucose and 1- ^{13}C labeled glucose.

R7: To explain their experiments investigating signaling pathways, the authors have proposed a model in which, in aggressive cells, glutamine activates STAT3 by both serine and tyrosine phosphorylation downstream of its presence (see Fig. 7). However, many of the connections that they try to make to explain this process appear to be potentially incompletely substantiated. For instance, while both serine phosphorylation of STAT3 and ERK1/2 phosphorylation appeared to occur concurrently following glutamine addition, the reviewer finds it difficult to conclude that glutamine mediates phosphorylation of STAT3 S727 via ERK1/2 without an experiment showing a reduction in this activity using some means to validate causality, such as showing a reduction in STAT3 S727 phosphorylation in response to an ERK inhibitor or a reduction in functional ERK levels. Unless such evidence to distinguish causation is available, it may be preferable to weaken claims such as the interactions shown between signaling proteins on the Fig. 7 schematic. (Again, it is possible that some of the references made when discussing this portion demonstrate such causal relations, but the citations in this section were linked to a numerical format that was incompatible with the subsequent bibliography. Even if this is the case, a more thorough discussion of how this causality was established should be given.)

A7: The reviewer brought out a very interesting point regarding involvement of Erk1/2. We performed additional experiments to understand if glutamine mediates phosphorylation of STAT3 S727 via ERK1/2. To explain the relationship between serine phosphorylation of STAT3 and ERK1/2 phosphorylation, we added PD9805, an inhibitor of MEK/ERK pathway. From the western blot results, we found that PD9805 efficiently blocked the ERK phosphorylation and also decreased serine phosphorylation level of STAT3. The result is shown below in C7 and included as Sup. Fig. 8.

We apologize for the confusion regarding references. In our revised manuscript, we have corrected the citations according to the MSB format. The reference below substantiates that ERK1/2 inhibition results in decreased STAT3 serine phosphorylation.

Chung J, Uchida E, Grammer TC, Blenis J (1997) STAT3 serine phosphorylation by ERK-dependent and -independent pathways negatively modulates its tyrosine phosphorylation. *Mol Cell Biol* **17**: 6508-6516

Gough DJ, Koetz L, Levy DE (2013) The MEK-ERK Pathway Is Necessary for Serine Phosphorylation of Mitochondrial STAT3 and Ras-Mediated Transformation. *PLoS One* 8

C7:

Page 26-27, paragraph 2: Deprivation of both Glc and Gln decreases tyrosine phosphorylation of STAT3, whereas Gln deprivation lead to decreased STAT3 serine phosphorylation levels through reduction in Erk 1/2 phosphorylation level; validated by addition of Erk1/2 phosphorylation inhibitor PD98059 (Sup. Fig. 8).

R8: In Fig. 5C, a protein called Stat3c is expressed, and in Fig. 5I, a compound called "Stattic" is used; these appear to respectively refer to a constitutively active form of STAT3 and an inhibitor of STAT3, as judging by the figure captions, but nowhere is this clearly stated. These conditions should be clearly defined in the relevant sections in the main text when they are discussed, as should all abbreviations or terms that are not widely known.

C8:

We thank the reviewer for identifying this error. In the revised manuscript, now we have clarified the difference between "Stat3c" and "Stattic" terms.

Page 22, paragraph 2: In line with these findings, inhibiting both STAT3 activity (through addition of static, a STAT3 inhibitor) and conversion of Gln into glutamate (through BPTES, a glutaminase inhibitor) results in a pronounced reduction of viability in aggressive cancer cells (Fig. 6I).

Page 45, Figure 6I caption: The addition of Stattic, inhibitor of STAT3, has a combinational cytotoxicity effect on both cell lines.

Page 21, paragraph 1: Significantly, when STAT3 is overexpressed in SKOV3 cells through transfecting with a constitutively active form of STAT3 (stat3c) we found that it rescues aggressive OVCA cells (SKOV3) from Gln deprivation induced reduction of cellular growth (Fig. 6C).

R9: This note may be minor, but on p. 23, the authors mention "In poorly aggressive OVCA cells, glutamine contributed more than 50% of pyruvate... [o]n the other hand, our results demonstrate that in aggressive OVCA cells, glutamine contributed to more than 50% of glutamate..." While it is clear that the authors intended the comparison in TCA cycle metabolite enrichment (excluded here) to show the differential contribution of glucose and glutamine to between each class of cell lines, it is not clear why they compared the glucose-to-pyruvate and

glutamine-to-glutamate contributions to one another between different lines. This should be further justified; otherwise, such a comparison should not be made without including additional information (e.g. aggressive cell glucose-to-pyruvate contribution and non-aggressive cell glutamine-to-glutamate contribution).

A9: We have reworded the sentence. We hope that this addresses the concern raised by the reviewer.

C9:

Page 25, paragraph 2: In both poorly-aggressive and aggressive OVCA cells, we found that Glc contributes more than 50% of pyruvate (M3 pyruvate and lactate was greater than 25% in 1:1 mixture of $^{13}\text{C}_6$ Glc and 1- ^{13}C labeled Glc). On the other hand, in aggressive OVCA cells, Gln contributes more than 50% of glutamate and 40% of TCA cycle fluxes, whereas in poorly aggressive OVCA cells Gln contributes around 15% of glutamate and TCA cycle fluxes.

R10: For any conclusions drawn regarding relative metabolic flux activity from labeling data, the observed system must be at isotopic steady state (i.e. labeling shouldn't change with time), but this did not appear to be discussed in the methods section. The authors should show that this does indeed hold for their system when they have drawn such conclusions (or at least cite references that establish or suggest that their conditions should be sufficient).

A10: We thank the reviewer for this suggestion and we have now cited the appropriate references in the methods section. Both "Glucose-Independent Glutamine Metabolism via TCA Cycling for Proliferation and Survival in B Cells " and "Beyond aerobic glycolysis: transformed cells can engage in glutamine metabolism that exceeds the requirement for protein and nucleotide synthesis" show that 24 hours is sufficient time for attaining steady state.

Le A, Lane AN, Hamaker M, Bose S, Gouw A, Barbi J, Tsukamoto T, Rojas CJ, Slusher BS, Zhang H, Zimmerman LJ, Liebler DC, Slebos RJ, Lorkiewicz PK, Higashi RM, Fan TW, Dang CV (2012) Glucose-independent glutamine metabolism via TCA cycling for proliferation and survival in B cells. *Cell Metab* **15**: 110-121

DeBerardinis RJ, Mancuso A, Daikhin E, Nissim I, Yudkoff M, Wehrli S, Thompson CB (2007) Beyond aerobic glycolysis: transformed cells can engage in glutamine metabolism that exceeds the requirement for protein and nucleotide synthesis. *Proc Natl Acad Sci U S A* **104**: 19345-19350

R11: It is unclear to the reviewer what the authors mean in the figures when the y-axis units on growth curves are given as "Relative Proliferation Normalized with RPMI." How is this term defined? Is the cell count on each given day divided by the corresponding cell count on the same day for cells grown in complete media? Terms such as this (including the normalized glucose uptake and lactate secretion rates in Fig. 6) need to be clearly stated.

A11: We have now corrected the units appropriately. To estimate relative proliferation, cell number on each given day in glutamine/glucose deprivation condition or other conditions is divided by the corresponding cell number on the same day for cells grown in complete medium without drugs.

C11:

Below are the corrections made in Figure 1, Figure 3, Figure 4, Figure 5, Figure 6, supplementary Figure 3, 4, and 7.

“Relative Proliferation Rate” is replaced by “Relative Proliferation”

“Relative Glucose Uptake Rate” is replaced by “normalized glucose uptake”

“Relative Lactate Secretion Rate” is replaced by “normalized lactate secretion”

“Relative Mitochondrial Membrane Potential” is replaced by “Normalized Mitochondrial Membrane Potential”

“Relative ATP Content” is replaced by “Normalized ATP content”

“Relative Reactive Oxygen Species” is replaced by “Normalized Reactive Oxygen Species

“Relative NADPH value” is replaced by “Normalized NADPH value”

“Relative Glutathione value” is replaced by “Normalized Glutathione value”

R12: As well, in the Fig. 3 caption, the sentence "OCR was normalized with value before injections" appears; again, it was unclear what this meant, and this normalization condition should be clearly defined.

A12: We have made this correction in the revised manuscript. “Relative OCR” is now replaced by “OCR was normalized with value before injection”

Here, the Relative OCR is the OCR value that is normalized with the OCR value before the first injection of Glutamine (time point 25 minutes) in Figure 2A.

R13: In Fig. 4J, the units on the y-axes are given as AU, which is acceptable, but what these arbitrary units are representing or are proportional to needs to be clearly stated. (Presumably, judging by the caption, this is cell number, but this should be shown explicitly somewhere, ideally on the figure.).

A13: We have now indicated in the figures what AU represents. In these figures, AU is proportional to cell number, which is now indicated in caption of Figure 4L.

C13:

Page 43, Figure 4L caption: Arbitrary Units (AU) is proportional to cell number.

R14: Loading control proteins should be provided on Figs. 5C-E, as they are for other Western Blots.

A14: We have provided the loading controls for Figs. 5C-E. We thank the reviewer for pointing out this omission.

R15: In estimating glycolytic/ETC pathway contribution to ATP synthesis using oligomycin, as was done for Fig. 2K, even if total ATP levels stay the same following oligomycin-mediated inhibition of ATP synthetase (as was shown), the notion that glycolysis will completely compensate for the disruption in respiration should only hold true if the total rate of ATP consumption remains unchanged, which seems difficult to validate. In using these measures to infer percentage contribution to ATP content, the authors should cite a source indicating that it is a valid measure as such; otherwise, they should label their data to more accurately reflect what was directly measured.

A15: We have cited the reference which motivated our experiment. Hao WS, Chang CPB, Tsao CC, Xu J (2010) Oligomycin-induced Bioenergetic Adaptation in Cancer Cells with Heterogeneous Bioenergetic Organization. *Journal of Biological Chemistry* **285**: 12647-12654. In this paper, authors added oligomycin for inhibiting OXPHOS activity. We agree that ATP consumption is difficult to measure and ATP content is what is generally measured. We found that with increasing concentrations of oligomycin there was no increase in lactate secretion thus indicating that OXPHOS inhibition is complete at the maximal concentration of oligomycin that we use. ATP content remains same with and without oligomycin, thus indicating that the ATP loss by inhibition of OXPHOS is compensated by the ATP gain through increase in glycolysis (reflected by the increase in lactate secretion). Therefore, we can calculate the ratio of OXPHOS over glycolysis by comparing the increased part of lactate secretion to the lactate secretion without oligomycin.

C15:

Page 10, paragraph 1: Therefore, the increased levels of glycolysis (indicated through lactate secretion) are likely a consequence of OXPHOS inhibition (Hao et al, 2010).

R16: In Fig. 3H, it is unclear why glucose-labeled and glutamine-labeled pyruvate and lactate are being compared to one another; these values measure completely different things (and would normally be expected to be very different), and indeed the differences between each condition appear to be very similar (with the minimum p-value for each). The purpose behind this figure and such a comparison should be better explained, and perhaps the statistical significance/p-value icons should be removed unless their purpose is otherwise necessary.

A16: We wanted to explain that pyruvate and lactate are mainly coming from glucose rather than glutamine. High glutamine to pyruvate flux will indicate high malic enzyme activity. However, our results indicate that this flux is quite low. We would like to mention that since both glucose and glutamine labeled medium had same medium composition, therefore, the total lactate and pyruvate concentrations should be same under these labeling conditions. This will allow direct comparison of M3 pyruvate and lactate from glucose and glutamine labeled medium. Based on the reviewer's suggestion, we have removed the p values since it is obvious that glucose is main source for pyruvate and lactate in each ovarian cancer cell lines.

R17: At some point in the manuscript (say either the main body or the supplemental figure captions), the authors should explain the significance of each of the protein levels and phosphorylation statuses examined in Fig. S5F is relevant. (E.g. The manuscript mentioned that some are indicating autophagy, but did not say which one(s).)

A17: Thank you for pointing out this omission. We now explain the significance of each protein level in the main body:

C17:

Page 22, paragraph 2: Furthermore, in aggressive OVCA cells, Gln/Glc deprivation results in increased phosphorylation levels of p38 MAPK, a protein kinase central to regulation of MAPK signaling pathway cascades; increased phosphorylation level of ACC, an enzyme integral to lipid metabolism regulated through AMPK signaling pathway; as well as decreased FAK phosphorylation levels for cellular adhesion. Additionally, Glc/Gln deprivation did not have significant effect on TSC2/GSK3 signaling, which regulates energy metabolism. However, Glc but not Gln deprivation enhanced autophagy in all cell lines indicated by increased LC 3 β -II and AKT phosphorylation, which regulate Glc metabolism in cancer cells (Sup. Fig. 6G).

R18: Although they don't need to include the descriptions in the methods and materials section, they authors should at least mention that the Seahorse, ATP measurements, ROS measurement assays, and all other protocols are given in the supplementary methods section (e.g. just one sentence in the methods and materials stating the additional protocols given in the supplemental section).

A18: We have now inserted the suggested sentence in Methods on page 30:

Additional protocols are given in the supplemental section.

R19: In the Supplementary Information section, the authors used a lot of unfamiliar abbreviations (e.g. TMRM, XTT) that were never clearly defined. The authors should define each abbreviation first before using it.

A19: We have defined all the abbreviations. Thank you.

C19: XTT and PMS:

Page 46, paragraph 4: Then wells were loaded with Tetramethylrhodamine, methyl ester (TMRM) in 37 °C for 15 min.

Page 46, paragraph 5: Cells were seeded in 96 well plates to 90% confluency. Fresh medium with H₂DCFDA was loaded in 37 °C for 30 minutes. Remove cell-permeant 2',7'-dichlorodihydrofluorescein diacetate (H₂DCFDA) medium and wash twice with PBS.

Page 47, paragraph 1: 66.6 μ l 1-methoxy N-methyl dibenzopyrazine methyl sulfate (PMS). PMS used was 0.5 mM in DMSO and 3.2 ml 2,3-Bis-(2-Methoxy-4-Nitro-5-Sulfophenyl)-2H-Tetrazolium-5-Carboxanilide (XTT, 251 mM in DMSO) were mixed to prepare XTT/1-methoxy PMS solution.

Page 47, paragraph 5: Following treatment with drugs and different media conditions cells were washed and then lysed for 20 minutes on ice with lysis buffer (50 mM HEPES pH 7.4, 150 mM NaCl, 1 mM ethylene glycol tetraacetic acid (EGTA), 10 mM sodium pyrophosphate, 100 mM NaF, 1.5 mM MgCl₂, 10%(v/v) glycerol, 1%(w/v) Triton X-100, protease inhibitor cocktail (Sigma), and phosphatase inhibitor (Sigma)).

R20: As indicated, the reviewer believes this manuscript should be published, but not until each of the previously-mentioned points is addressed.

A20: We thank the reviewer for the positive comments and critical reading of the manuscript. We feel that because of these modifications the quality of our manuscript has been greatly improved.

Reviewer #2:

R1 : In this manuscript Yang et al showed that ovarian cancer cells have distinctive glutamine metabolic traits which correlate with tumor's invasive capacity. The study is original, and the study of glutamine metabolism is comprehensive. However, the relation between metabolism and aggressiveness of the tumor remains speculative and should not be emphasized in the title and in the text. In particular, the assumption that the in-vitro invasive capacity of the cells, assessed by a transwell migration assay, is readout for aggressiveness of the tumor is not supported by experimental evidences. An in-vivo approach would help clarifying this basic point and strengthen the study.

A1 : We thank the reviewer for encouraging us to perform in vivo experiments. As suggested, we have performed in vivo studies to demonstrate the effect of inhibiting glutaminase in mice with tumors grown from IGROV1 (a poorly aggressive ovarian cancer cell line) and SKOV3ip1 (aggressive ovarian cancer cell line). A highlight of our *in vivo* experiment is the discovery that inhibition of glutaminolysis is more detrimental to aggressive OVCA cells than to their nonaggressive counterparts. GLS1-targeted siRNA significantly decreased cancer growth and invasion in mice bearing ovarian tumors derived from glutamine dependent (aggressive, SKOV3ip1) as compared to glutamine independent (poorly aggressive, IGROV1) cells. Please see below for the corrected text and new figures added (Fig. 5B-5H). We hope that through both *in vitro* mechanistic studies and *in vivo* therapeutics-targeting experiments we have clearly demonstrated that aggressive OVCA cells are markedly dependent on Gln, whereas poorly aggressive OVCA cells are Gln-independent.

C1:

Page 18, paragraph 3: Next, we examined the biological effects of GLS1 siRNA in Gln-independent (IGROV1) and Gln-dependent (SKOV3ip1) in mouse models. Before initiating *in vivo* experiments, we checked the efficacy of siRNA for silencing GLS1 in SKOV3ip1 cells using targeted GLS1 siRNA. Transfection of SKOV3ip1 cells with the targeted siRNA resulted in >70% decrease in GLS1 mRNA levels (Fig. 5B). Trypan blue dye assay confirmed that transfection of cells with siRNAs did not affect cell viability after 48 hours of transfection, suggesting that siRNA was not toxic to cells. siRNA were incorporated into neutral liposome DOPC (1,2-Dioleoyl-sn-Glycero-3-phosphatidylcholine). For therapy experiments, siRNA administration was started 1 week after tumor cell injection subcutaneously. Mice were divided into the following 2 groups (n = 5 mice per group): a) control siRNA-DOPC, and b) GLS1 siRNA-DOPC. All of the animals were sacrificed two weeks after inoculation. As seen in Fig. 5C-5E, treatment with GLS1 siRNA-DOPC in Gln independent IGROV1 tumor bearing mice did not affect tumor volume or invasive phenotype compared to control siRNA treated group. In contrast, GLS1 gene silencing in Gln dependent SKOV3ip1 tumor-bearing animals showed significant reduction in both tumor weight (60%; p = 0.007), tumor volume (75%; p = 0.008) and invasive phenotype compared to control siRNA-DOPC treated groups (Fig. 5F-5H).

Figure 5

R2 : To support the results obtained with the cell lines, the authors analyzed transcriptional profiles of ovarian tumor samples. They concluded that the ratio between GLUD1 and GLUL can predict patients' prognosis. However, it is known that the expression of glutamine metabolic genes can be regulated post-translationally, in particular, Glutamine Synthetase protein abundance is controlled by protein degradation in many tissues. Thus the potential of these biomarkers for aggressiveness should be validated at the protein levels in the cell lines, and more importantly, in samples from ovarian tumor tissues. The authors should also comment on the opposite 'hazard ratios' found for genes of the glutaminolytic pathway (GLS and GLUD) and the glutamine transporters (SLC1A5, SLC38A1). This is in particularly unclear as the glutamine transporters feed the reactions catalyzed by the glutaminolytic enzymes (Fig 1G).

A2 : We thank the reviewer for the insightful comments. As suggested, we have performed Western blots to estimate protein level for glutaminase and glutamine synthetase in ovarian cancer cell lines (see new Figure 4B). As seen below, the protein expression of GLUL

(glutamine synthetase) is decreased in highly aggressive OVCA cells. In contrast, protein expression of GLS1 (glutaminase) is increased in highly aggressive cells. Importantly, protein expression results were in agreement with gene expression results obtained from Kyotov38 and support our conclusions. We followed the suggestion of the reviewer to check our findings using ovarian tumor tissues. We estimated the protein expression (through Immunohistochemistry) of GLS1 and GLUL using the tissue microarrays (TMA). The protein expression scores (please see methods for scoring method) obtained from TMA (epithelial ovarian cancer patient samples (n= 139)) were correlated to survival outcome. Our results as seen in Figs 5A-5C clearly validate our findings obtained from the TCGA data. Please see below for the added text in the manuscript. TMA data is shown above in Fig 5 pasted as answer to C1 comment. We agree that glutamine transporter genes have low hazard ratios, but the glutaminolysis pathway has high hazard ratio so there remains the possibility for other glutamine import systems are being upregulated. Moreover, few studies (please see reference below) have shown that a portion of glutamine taken up through SLC1A5 is rapidly exported through the bidirectional amino acid transporter SLC7A5 in exchange for uptake of extracellular essential amino acids. Thus, the network study between glutamine importers and transporters along with other essential amino acids may address this discrepancy. However, we believe that our TMA and cell line westerns validate our observations.

David R. Wise, Craig B. Thompson. Glutamine Addiction: A New Therapeutic Target in Cancer. Trends Biochem Sci. 2010 August; 35(8): 427–433.

C2:

Page 18, paragraph 2: Clinical significance of glutamine catabolism and therapeutic effectiveness of GLS1 siRNA in ovarian cancer models

To further evaluate the clinical significance of the ratio of Gln catabolism over anabolism, we assessed GLS1 and GLUL protein expression in epithelial ovarian cancer samples (n= 139) using tissue microarrays (TMA) and correlated it to survival outcome. High GLS1 protein expression was associated with worse overall survival, whereas high GLUL protein expression was associated with better survival (Fig. 5A). In concordance with our previous results obtained from gene expression data from TCGA, we found that the ratio of Gln catabolism over

anabolism (GLS1/GLUL) through protein expression was also significantly related to worse overall survival ($p < 0.001$).

R3 : One important factor which contributes to tumor aggressiveness is the cell proliferative potential. The results presented in this study do not actually compare the proliferation rates of the cell lines, only the decrease in proliferation under starved conditions. The possible correlation between the basal proliferation rate of the cells and the invasive ability should be assessed. It is possible that higher proliferation rate leads to both, higher dependency on glutamine and higher invasive potential; hence, the correlation between glutamine dependency and invasiveness is secondary to the proliferation rate of the cells.

A3 : We appreciate that the reviewer has suggested an alternative factor for correlating our observations. As suggested, we have performed the growth rate experiments for all cell lines, and found that their growth rate did not correlate with invasive capacity.

This result didn't surprise us because it has been previously observed that under invasive or metastatic conditions, cancer cells may have lower proliferation. Please see the references cited below. Hence, we believe that invasive phenotype is more closely correlated to glutamine addiction. Nevertheless, the reviewer brings forth an interesting point which may warrant a separate study focusing on nutrient utilization, protein synthesis, growth rate using stable isotope analysis.

Evdokimova V, Tognon C, Ng T, Sorensen PH (2009) Reduced proliferation and enhanced migration: two sides of the same coin? Molecular mechanisms of metastatic progression by YB-1. *Cell Cycle* **8**: 2901-2906

Hoek KS, Eichhoff OM, Schlegel NC, Dobbeling U, Kobert N, Schaerer L, Hemmi S, Dummer R (2008) In vivo switching of human melanoma cells between proliferative and invasive states. *Cancer Research* **68**: 650-656

Kallioniemi A (2012) Bone morphogenetic protein 4-a fascinating regulator of cancer cell behavior. *Cancer Genet* **205**: 267-277

Svensson S, Nilsson K, Ringberg A, Landberg G (2003) Invade or proliferate? Two contrasting events in malignant behavior governed by p16(INK4a) and an intact Rb pathway illustrated by a model system of basal cell carcinoma. *Cancer Res* **63**: 1737-1742

R4: The study characterized the metabolic fate of glutamine and glucose carbons in representative cell lines (Fig 2 and 3). In this context, the interpretative advantage of using a mixture of U-¹³C₆ and 1-¹³C glucose is questionable and more likely to generate confusion due to isomer mixing from different sources. Moreover the activity of malic enzyme (as shown in Fig 3) which produces glutamine-derived ¹³C₃ pyruvate is not taken into account in Figure 2. This affects the potential isotopomer distribution represented in scheme 2E and the conclusions thereafter.

A4 : We have used 1:1 mixture of U-¹³C₆ glucose and 1-¹³C₁ glucose to calculate oxidative PPP fluxes. The 1-¹³C labeled glucose loses the labeled carbon through oxidative PPP, and is converted to generate M0 pyruvate. However, when 1-¹³C labeled glucose is converted into pyruvate through glycolysis, it generates M0 pyruvate and M1 pyruvate. Therefore, we can calculate the difference between the percentage of M0 pyruvate and M1 pyruvate (M0 pyruvate %-M1 pyruvate%) to represent the relative oxidative PPP fluxes. This approach has been used in several recent ¹³C based isotope studies including the one shown below.

Gaglio D, Metallo CM, Gameiro PA, Hiller K, Danna LS, Balestrieri C, Alberghina L, Stephanopoulos G, Chiaradonna F (2011) Oncogenic K-Ras decouples glucose and glutamine metabolism to support cancer cell growth. *Mol Syst Biol* 7: 523

The reason that we do not take malic enzyme into consideration is because the flux is too low (see figure 3H). Since the glutamine's contribution to pyruvate is very low, therefore, it will not affect the metabolite isotopomer distribution. Furthermore, in Figure 2F and 2G, we only compare one turn of TCA cycle when glutamine enters into TCA cycle through oxidative TCA (M4 fumarate) and reductive TCA (M5 citrate) pathways.

C4:

Page 13, paragraph 2: In contrast, poorly aggressive cancer cells had increased oxidative pentose phosphate pathway flux (PPP) compared to aggressive cancer cells (Fig. 3J), which was estimated using M0 and M1 labeling of lactate in cells cultured with 1:1 mixture of U-¹³C₆ Glc and 1-¹³C labeled Glc. 1-¹³C labeled Glc loses the labeled carbon when it is routed through oxidative PPP, and then is converted into M0 pyruvate (Gaglio et al, 2011).

R5 : The authors refer to the exchange of metabolites between cells and the medium (Fig 2 and 6) as uptake/secretion rate. However, the time unit for this rate is not shown. Furthermore, the normalization of the exchange rates suffers a more fundamental problem: the consumption/secretion rates should take into account the cell growth during the whole experimental period assessed and not just the number of cells at the beginning or end of the study. It is intuitive that faster growing cells consume more nutrients per unit of time than slow ones, independently of their metabolic traits. For this reason, a normalization that takes into account cell growth is necessary to evaluate potential differences in the metabolic phenotype. Without this, the conclusions drawn from such a study are invalid.

A5: The time unit is 24 hours. Since most of the metabolites attain steady state within this time we use it to measure fluxes. Different normalization methods can be used (protein, cell number and dna content). The aforementioned methods have been widely used in metabolic flux analysis studies. We have applied all the three for these cells and have not found any major differences. As reviewer acknowledges that these fluxes are hard to estimate, the only solution is to estimate them at multiple time points and for shorter durations. However, that becomes

prohibitively expensive and large scale if various drugs and conditions are involved. We believe that our results strongly support our main hypothesis and conclusion. We have followed the currently utilized paradigm for normalization throughout the manuscript. Below are few references where similar methods were adopted.

Gaglio D, Metallo CM, Gameiro PA, Hiller K, Danna LS, Balestrieri C, Alberghina L, Stephanopoulos G, Chiaradonna F (2011) Oncogenic K-Ras decouples glucose and glutamine metabolism to support cancer cell growth. *Mol Syst Biol* 7: 523

Metallo, Christian M., Paulo A. Gameiro, Eric L. Bell, Katherine R. Mattaini, Juanjuan Yang, Karsten Hiller, Christopher M. Jewell, et al. "Reductive Glutamine Metabolism by IDH1 Mediates Lipogenesis under Hypoxia." *Nature* 481, no. 7381 (January 19, 2012): 380–384.

R6 : In figure 4H, the authors measured mitochondrial respiration in the "glutamine independent" cells in the absence of glutamine and presence of MSO, an inhibitor of Glutamine Synthetase. The results of this experiment are misinterpreted: the decrease in OCR observed upon MSO addition cannot be justified by a supposed decrease in the flux of glutamine into the TCA cycle. In fact, by inhibiting the conversion of glutamate to glutamine, MSO is expected to increase the availability of glutamate for the TCA cycle. The same principle applies for the effect of MSO on glutamine-fed cells, where MSO stimulated cell invasiveness, potentially by favoring glutamate flux into the TCA cycle hence, de-facto increasing glutaminolysis. This however, needs to be assessed by direct tracing of glutamine under MSO-treated conditions.

A6 : We thank the reviewer for this comment. We would like to clarify that decrease in OCR, with the addition of MSO is due to the reduction in glucose's conversion into glutamine through the TCA cycle. Since the medium did not contain glutamine at the start of measurement (MSO injection), the decrease in OCR could not have been from exogenous glutamine. The addition of MSO decreases glucose's conversion into glutamine through TCA cycle; therefore, there is a decrease of OCR. As seen in the figure, when we add glutamine, it largely increases Gln's entry into TCA cycle (glutaminolysis) and thus increases TCA activity.

As suggested by the reviewer, we performed glutamine tracer experiment under MSO conditions. As seen below through glutamine labeling experiments, MSO increases Gln's entry into TCA cycle in poorly aggressive OVCA cells. We have included this new data in the main text. Please see the correction below.

C6:

Addition of Figure 4K & Page 18, paragraph 1: Remarkably, inhibiting Gln synthesis increases aggressiveness in poorly aggressive OVCA cells (Fig. 4J). Using U-13C5 Gln labeling experiments we found that under MSO conditions, OVCA cells had increased conversion of Gln into TCA metabolites to support cancer metastasis (Fig 4K). There was significant increase in fluxes of citrate, α -KG, and other TCA metabolites which are directly derived from U-13C5 Gln (Fig 4K).

R7 : It would be of interest to assess whether STAT3 phosphorylation is specifically regulated by glutamine or whether it can be reproduced by other non-essential amino acids.

A7 : We thank the reviewer for this interesting suggestion. We found that glycine, serine, aspartate, asparagine and various combinations of their deprivations did not alter STAT3 phosphorylation (see the supplementary figure 6F). Please see the figure below and added corrections.

C7:

Page 21, paragraph 1: The deprivation of other amino acids, such as serine, glycine, asparagines, and aspartate, did not affect the STAT3 phosphorylation in OVCA cells (Sup. Fig. 6F).

Page 49, Supplementary Figure 6F caption: The effect of serine, glycine, asparagine, and aspartate deprivations on STAT3 phosphorylation. Cell culture medium is MEM and was adjusted to similar nutrients component as RPMI. And different amino acids are deprived

Ser	+	-	-	+	+	+	-	+
Gly	+	-	+	-	+	+	-	+
Asn	+	-	+	+	-	+	+	-
Asp	+	-	+	+	+	-	+	-

R8 : The authors hypothesized that the increase in serine uptake observed during glucose deprivation, could produce glycolytic intermediates. It is unclear (and unexplained) how serine may be utilized by the cells under glucose-starved conditions and the references used by the authors do not provide such explanation.

A8: We thank the author for highlighting this error. We meant that serine and glycine could serve as alternative energy sources for cell survival. We have made the correction in the aforementioned statement.

C8:

Page 24, paragraph 1: This may be due to the absence of Glc mediated serine synthesis for the SOG (serine, one-carbon cycle, glycine synthesis) pathway to generate NADPH, ATP and purines. Therefore, cancer cells will uptake more exogenous serine to meet their nutritional requirement for survival (Tedeschi et al, 2013, Maddocks et al, 2013).

2nd Editorial Decision

12 March 2014

Thank you again for submitting your work to Molecular Systems Biology. We have now heard back from the two referees who agreed to evaluate your manuscript. As you will see from the reports below, both referees think that their main concerns have been satisfactorily addressed. However, reviewer #2 lists two issues that we would ask you to address in a revision of this work. As this reviewer suggests, we would recommend avoiding (or very cautiously using, when appropriate) the term "aggressiveness" throughout the manuscript. Moreover, we would like to ask you to describe in more detail the "pattern of invasion" for the two OVCA cell lines shown in Fig. 5E and 5H.

Reviewer #1:

The reviewer appreciates the comprehensive revision of the manuscript and the point-by-point response to each of the aforementioned comments. The authors' efforts have improved the manuscript's clarity and comprehensibility tremendously, and the additional experiments and figures have helped resolve a number of the remaining questions. The reviewer feels that the conclusions are now well-substantiated and presented effectively. Aside from one minor, easily-correctable potential error (in the caption of Figure 5B, "miRNA" presumably refers to "mRNA"), the manuscript appears ready for publication in the reviewer's eyes.

Reviewer #2:

In the new manuscript the authors have significantly improved the work and answered many of my concerns. There are still a couple of issues that needs to be clarified/addressed.

1. One of my major concerns with the previous manuscript is the definition of aggressiveness and its relationship to glutamine metabolism. The authors now provided a link between GLS1 expression and survival of patients. Furthermore, they demonstrated in an in-vivo study that the glutamine dependent and invasive cells (SKOV3ip1) were more affected by GLS1 knockdown as compared to glutamine independent and non-invasive cells (IGROV1). However, the tumor's size and volume of IGROV1 were much higher than those of SKOV3ip1 (Figure 5C-H), questioning the definition of aggressiveness. There were no invasive or survival studies in these mice, and hence, one cannot draw the conclusion that glutaminolysis is linked to aggressiveness from this study. I would therefore request to tone down the use of this term in the title and in the text.

2. In Figure 4-I-L, the authors discussed the use of MSO as a way to stimulate glutaminolysis and with that invasiveness. While the effect of MSO on invasiveness is clear, the mechanism is certainly not obvious. From the new carbon-tracing experiment with ¹³C-Glutamine, I am not convinced that MSO stimulated glutaminolysis, but it only increased the steady-state levels of TCA cycle metabolites (by blocking cataplerosis?). I also still cannot understand why by preventing exit of TCA cycle metabolites to form glutamine, MSO reduced respiration in glutamine-starved cells. Overall, I think the mechanism of MSO is unclear and confusing and should be either mechanistically explained or removed from the manuscript.

2nd Revision - authors' response

18 March 2014

Response to Editor's comments

Thank you again for submitting your work to Molecular Systems Biology. We have now heard back from the two referees who agreed to evaluate your manuscript. As you will see from the reports below, both referees think that their main concerns have been satisfactorily addressed. However, reviewer #2 lists two issues that we would ask you to address in a revision of this work. As this reviewer suggests, we would recommend avoiding (or very cautiously using, when appropriate) the term "aggressiveness" throughout the manuscript.

We thank you and the reviewers for evaluating our manuscript again and giving us positive comments and assessment. We feel that with the suggested changes, the manuscript's clarity has been significantly improved and it will remove any ambiguity that was present in the previous version. As suggested by you and the 2nd reviewer, we have avoided the term "aggressiveness" in our revised manuscript. We have removed it from the title and throughout the manuscript have replaced it with invasiveness. Thus, we are now referring "aggressive" and "poorly aggressive" cells by "high-invasive" and "low-invasive", respectively.

Moreover, we would like to ask you to describe in more detail the "pattern of invasion" for the two OVCA cell lines shown in Fig. 5E and 5H.

We thank you for this suggestion and have revised our description. In our revised manuscript, on Page 18, our detailed description is included below:

As seen in Fig. 5C-5E, treatment with GLS1 siRNA-DOPC in Gln independent IGROV1 tumor bearing mice did not affect tumor volume or deep tumor infiltration through the muscle layers compared to control siRNA treatment. In contrast, GLS1 gene silencing in Gln dependent SKOV3ip1 tumor-bearing animals showed significant reduction in both tumor weight (60%; $p = 0.007$), and tumor volume (75%; $p = 0.008$) when compared to control siRNA-DOPC treated groups. Moreover, GLS1 siRNA-DOPC blocked tumor infiltration into surrounding tissues (Fig. 5F-5H).

On a more editorial level, we would like to ask you to provide individual files for all Supplementary Figures. Each file should contain the Supplementary Figure and the corresponding Supplementary Figure legend.

As per the request, we have included individual files for all supplementary figures and tables. We have included the legend of the supplementary figures in the main text after the legends of the figures.

Please resubmit your revised manuscript online, with a covering letter listing amendments and responses to each point raised by the referees. Please resubmit the paper ****within one month**** and ideally as soon as possible. If we do not receive the revised manuscript within this time period, the file might be closed and any subsequent resubmission would be treated as a new manuscript. Please use the Manuscript Number (above) in all correspondence.

We thank the editor, Dr. Polychronidou, for assessing our revised manuscript and for her valuable feedback. We appreciate the editors' efforts in providing us with a clearer focus of our manuscript's scientific intent. We feel that manuscript has substantially improved after the suggested changes.

Legend: R: reviewer's comment, A: author's response, C: Corrections

Response to Reviewer's comments

Reviewer #1

R1: The reviewer appreciates the comprehensive revision of the manuscript and the point-by-point response to each of the aforementioned comments. The authors' efforts have improved the manuscript's clarity and comprehensibility tremendously, and the additional experiments and figures have helped resolve a number of the remaining questions. The reviewer feels that the conclusions are now well-substantiated and presented effectively. Aside from one minor, easily-correctable potential error (in the caption of Figure 5B, "miRNA" presumably refers to "mRNA"), the manuscript appears ready for publication in the reviewer's eyes.

A1: We thank the reviewer for insightful comments and appreciating our work. We thank the reviewer for identifying the error in figure caption 5B. We have changed the word "miRNA" to "mRNA".

Reviewer #2:

R1: In the new manuscript the authors have significantly improved the work and answered many of my concerns. There are still a couple of issues that needs to be clarified/addressed. One of my major concerns with the previous manuscript is the definition of aggressiveness and its relationship to glutamine metabolism. The authors now provided a link between GLS1 expression and survival of patients. Furthermore, they demonstrated in an in-vivo study that the glutamine dependent and invasive cells (SKOV3ip1) were more affected by GLS1 knockdown as compared to glutamine independent and non-invasive cells (IGROV1). However, the tumor's size and volume of IGROV1 were much higher than those of SKOV3ip1 (Figure 5C-H), questioning the definition of aggressiveness. There were no invasive or survival studies in these mice, and hence, one cannot draw the conclusion that glutaminolysis is linked to aggressiveness from this study. I would therefore request to tone down the use of this term in the title and in the text.

A1: We greatly appreciate the reviewer's useful suggestions. As recommended, we have avoided the term "aggressiveness" in our revised manuscript. We have removed "aggressive" from the title and throughout the manuscript have replaced it with invasiveness. In our revised manuscript, we refer "aggressive" and "poorly aggressive" cells as "high-invasive" and "low-invasive", respectively.

R2: In Figure 4-I-L, the authors discussed the use of MSO as a way to stimulate glutaminolysis and with that invasiveness. While the effect of MSO on invasiveness is clear, the mechanism is certainly not obvious. From the new carbon-tracing experiment with ¹³C-Glutamine, I am not

convinced that MSO stimulated glutaminolysis, but it only increased the steady-state levels of TCA cycle metabolites (by blocking cataplerosis?). I also still cannot understand why by preventing exit of TCA cycle metabolites to form glutamine, MSO reduced respiration in glutamine-starved cells. Overall, I think the mechanism of MSO is unclear and confusing and should be either mechanistically explained or removed from the manuscript.

A2: Based on reviewer's suggestion, we have removed figures related to MSO's effect on glutaminolysis. In our previous revision, we followed reviewer's suggestion for performing isotope tracing experiments. Although we did see increase in concentrations of TCA cycle metabolites, we agree with the reviewer that MSO's effect on cancer cells' glutamine metabolism needs a detailed mechanistic study. Hence, we have removed MSO data pertaining to metabolism. Thank you for the critical review of our results.